# Transcriptional licensing is required for Pyrin inflammasome activation in human macrophages and bypassed by mutations causing familial Mediterranean fever

Matthew S. J. Mangan[1,2]*, Friederike Gorki[1☉], Karoline Krause[3,4☉], Alexander Heinz[5], Anne Pankow[6], Thomas Ebert[7], Dieter Jahn[8,9], Karsten Hiller[5,10], Veit Hornung[7], Marcus Maurer[3,4], Florian I. Schmidt[1], Ralf Gerhard[11], Eicke Latz[1,2,12,13]*

1 Institute of Innate Immunity, University Hospital Bonn, University of Bonn, Bonn, Germany, 2 German Center for Neurodegenerative Diseases, Bonn, Germany, 3 Institute of Allergology, Charité–Universitätsmedizin Berlin, corporate member of Freie Universität Berlin and Humboldt-Universität zu Berlin, Berlin, Germany, 4 Fraunhofer Institute for Translational Medicine and Pharmacology ITMP, Allergology and Immunology, Berlin, Germany, 5 Department of Bioinformatics and Biochemistry, Braunschweig Integrated Center of Systems Biology (BRICS), Technische Universität Braunschweig, Braunschweig, Germany, 6 Medizinische Klinik mit Schwerpunkt Rheumatologie und Klinische Immunologie AG Digitale Medizin in der Rheumatologie/ Rheumatologie 4.0 Charité—Universitätsmedizin Berlin (CCM), Berlin, Germany, 7 Gene Center and Department of Biochemistry, Ludwig-Maximilians-Universität München, Munich, Germany, 8 Institute for Microbiology, Technische Universität Braunschweig, Braunschweig, Germany, 9 Braunschweig Integrated Center of Systems Biology (BRICS), Technische Universität Braunschweig, Braunschweig, Germany, 10 Computational Biology of Infection Research, Helmholtz Centre for Infection Research, Braunschweig, Germany, 11 Institute of Toxicology, Hannover Medical School, Hannover, Germany, 12 Department of Infectious Diseases & Immunology, UMass Medical School, Worcester, Massachusetts, United States of America, 13 Centre of Molecular Inflammation Research, Norwegian University of Science and Technology, Trondheim, Norway

☉ These authors contributed equally to this work.
* matthew.mangan@ukbonn.de (MSJM); eicke.latz@uni-bonn.de (EL)

**Data Availability Statement:** All relevant data are within the paper and its Supporting Information files.

## Abstract

Pyrin is a cytosolic immune sensor that nucleates an inflammasome in response to inhibition of RhoA by bacterial virulence factors, triggering the release of inflammatory cytokines, including IL-1β. Gain-of-function mutations in the *MEFV* gene encoding Pyrin cause autoinflammatory disorders, such as familial Mediterranean fever (FMF) and Pyrin-associated autoinflammation with neutrophilic dermatosis (PAAND). To precisely define the role of Pyrin in pathogen detection in human immune cells, we compared initiation and regulation of the Pyrin inflammasome response in monocyte-derived macrophages (hMDM). Unlike human monocytes and murine macrophages, we determined that hMDM failed to activate Pyrin in response to known Pyrin activators *Clostridioides difficile* (*C. difficile*) toxins A or B (TcdA or TcdB), as well as the bile acid analogue BAA-473. The Pyrin inflammasome response was enabled in hMDM by prolonged priming with either LPS or type I or II interferons and required an increase in Pyrin expression. Notably, FMF mutations lifted the requirement for prolonged priming for Pyrin activation in hMDM, enabling Pyrin activation in the absence of additional inflammatory signals. Unexpectedly, in the absence of a Pyrin response, we found that TcdB activated the NLRP3 inflammasome in hMDM. These data

**Funding:** This work was funded by the Deutsche Forschungsgemeinschaft (DFG, German Research Foundation) – Project-ID 369799452 (TRR237) (E. L.), Project-ID 414786233 (SFB1403) (E.L.), Project ID 432325352 (SFB1454) (E.L.), DFG (GE1017/5-1) (R.G) and Germany's Excellence Strategy – ImmunoSensation2 – Project-ID 390873048 (EXC2151) (E.L.), The work was also supported by the Helmholtz Gemeinschaft, Zukunftsthema 'Immunology and Inflammation' (ZT-0027) (E.L.) and the German-Israeli Foundation for Scientific Research and Development, Grant No: 1085 (V.H.) This project was partially funded by the Niedersächsiches Ministerium für Wissenschaft und Kultur (MWK), Niedersächsisches Vorab [grant number ZN3380] (K.H.). The funders had no role in the study design, data collection and analysis, decision to publish, or preparation of the manuscript.

**Competing interests:** The authors have declared that no competing interests exist.

**Abbreviations:** BMDM, bone marrow–derived macrophage; FMF, familial Mediterranean fever; GTD, glucosyltransferase domain; hPyrin, human Pyrin; KO, knockout; LDH, lactate dehydrogenase; LPS, lipopolysaccharide; MOI, multiple of infection; mPyrin, murine Pyrin; PA, protective antigen; PAAND, Pyrin-associated autoinflammation with neutrophilic dermatosis; PBMC, peripheral blood mononuclear cell; PM, peritoneal macrophage; siRNA, small interfering RNA; PSTPIP1, protein proline serine threonine phosphatase-interacting protein 1; TLR, Toll-like receptor; WT, wild-type.

demonstrate that regulation of Pyrin activation in hMDM diverges from monocytes and highlights its dysregulation in FMF.

## Introduction

Inflammasome-initiating proteins are cytosolic sensors that mediate a posttranslational inflammatory response to pathogens or cell stress. Upon detecting a stimulus, these sensors recruit the adapter protein ASC, enabling activation of caspase-1. Active caspase-1 then mediates cleavage and release of pro-inflammatory cytokines including IL-1β, and pro-inflammatory cell death by cleaving the pore-forming molecule gasdermin D [1]. Notably, rather than relying solely on direct detection of pathogens, some inflammasome sensors have instead evolved to detect infection or cellular stress by monitoring disruption of cellular homeostasis [2]. A leading example of this is the Pyrin inflammasome, encoded by the *MEFV* gene, which is activated in response to inhibition of RhoA [3]. This small G protein controls cytoskeletal rearrangement and is essential for immune cell migration and phagocytosis, among other functions [4]. Unsurprisingly, given its role in fundamental cellular processes, RhoA is a target of numerous bacterial virulence factors from pathogenic bacteria, including *Clostridioides difficile* toxins A and B (TcdA and TcdB, respectively) [5]. Another prominent sensor of homeostasis is the NLRP3 inflammasome, which detects a wide range of events that converge to cause either mitochondrial dysfunction or loss of osmotic control of the cytosol [6]. Interestingly NLRP3 is also activated by perturbation of RhoGTPase family members, as the CNF-1 toxin, which modifies and permanently activates Rac, triggers NLRP3 activation [7].

Due to their high inflammatory potential, inflammasome forming sensors are strictly regulated by posttranslational controls. Under resting conditions, Pyrin is maintained in an inactivate state by 2 distinct mechanisms regulated by RhoA signaling. The most well characterized of these is phosphorylation of Pyrin at residues Ser208 and Ser242 by PKN1/2, members of the PKC superfamily [8–11]. Phosphorylation of Pyrin enables the subsequent binding of 14-3-3 proteins, which sequesters Pyrin in an inactive state. The second mechanism regulating Pyrin activation is less well understood. Pretreatment of cells with colchicine, a microtubule destabilization agent, inhibits Pyrin activation but does not prevent dephosphorylation of Pyrin, demonstrating that these 2 regulatory mechanisms are mutually exclusive [10]. Interestingly, human and mouse Pyrin share these regulatory mechanisms, although murine Pyrin does not contain the C-terminal B30.2 domain due to a frameshift mutation [12].

Understanding the regulatory mechanisms governing Pyrin is particularly important as mutations in the *MEFV* gene cause the hereditary autoinflammatory disorder, familial Mediterranean fever (FMF). FMF is characterized by recurrent attacks of fever, serositis, and abdominal pain and can, over time, cause secondary AA amyloidosis, leading to kidney failure [13]. Mutations linked to FMF are mostly amino acid substitutions in the C-terminal B30.2 domain [14]. Though the function of the B30.2 domain is still relatively unclear, these mutations are suggested to perturb the colchicine-dependent regulatory mechanism of Pyrin. This is supported by studies showing that Pyrin variants containing FMF mutations are resistant to inhibition by colchicine and that dephosphorylation is sufficient to activate the FMF form of Pyrin [15,16]. However, as the B30.2 domain has been lost in the mouse, it is difficult to model this disease. Thus, there is a need for further research into the effects of these mutations in human cell types to understand how they alter Pyrin regulation.

*C. difficile* infection is a leading cause of hospital-associated mortality through diarrhea and pseudomembranous colitis triggered by antibiotic therapy-mediated dysbiosis. All of these effects are entirely dependent on the expression of TcdA and TcdB, although TcdB is sufficient to cause disease [17]. Yet, the role of the inflammasome in the immune response to *C. difficile* infection is controversial. In vitro, treatment of bone marrow–derived macrophages (BMDMs) with relatively high concentrations of TcdA or TcdB triggered IL-1β secretion that was completely ablated in ASC KO BMDM, while only a minor reduction was observed in NLRP3 KO BMDM [18]. The inflammasome response in BMDM to both these toxins was subsequently demonstrated to be dependent on the Pyrin inflammasome [3]. In the human macrophage line, THP1, a compound shown previously to inhibit NLRP3, glyburide, could inhibit TcdB-mediated IL-1β secretion [18]. However, there was no genetic confirmation of the result and it is not clear how specific glyburide is for NLRP3. In vivo experiments in ASC KO mice examining the effect of TcdA and TcdB directly, rather than through *C. difficile* infection, demonstrated that inflammasome activation in response to these toxins increased tissue damage in an IL-1β-dependent manner [18]. A subsequent study demonstrated a role for caspase-1 in controlling *C. difficile* infection [19]. Experiments in both NLRP3 KO and Pyrin KO mice found that neither sensor impacted the severity of the disease [19,20]. It is conceivable that the pathology of the human pathogen *C. difficile* is not fully recapitulated in the murine model. Therefore, investigations into the inflammasome response to *C. difficile* in human cells are important and have so far included peripheral blood mononuclear cells (PBMCs), monocytes, and neutrophils [12]. In this study, we assessed the inflammasome response to *C. difficile* and its toxins, TcdA and TcdB, in a human macrophage model using M-CSF monocyte-derived macrophages.

## Results

### Inflammasome activation by *C. difficile* in hMDM is dependent on the expression of its toxins and can be blocked by NLRP3 inflammasome inhibition

To investigate whether the secreted virulence factors from *C. difficile* could elicit an inflammasome response in primary hMDM, we treated the cells with conditioned supernatant from *C. difficile*. As production of TcdA and TcdB is crucial for both *C. difficile*-driven pathology and the activation of the Pyrin inflammasome, 2 different strains of *C. difficile* were used: one proficient for production of both toxins A and B, and one deficient for both. We also assessed the effect of prior exposure to Toll-like receptor (TLR) ligands on the inflammasome response by priming one hMDM group with lipopolysaccharide (LPS) before exposure to the supernatant from *C.difficile*. In addition, we preincubated 1 group of cells with the potent and highly specific NLRP3 inhibitor CP-456,773 (also known as CRID3 or MCC950) [21] to determine if the response was NLRP3 dependent.

We determined that *C. difficile*-conditioned supernatant induced the release of 2 inflammasome-dependent cytokines, IL-1β and IL-18, from hMDM. The cytokine release was entirely toxin dependent in the cells treated with the conditioned supernatant (Fig 1A), though this data is correlative as the strains are not isogenic. Notably, IL-18, which is transcribed independent of TLR stimulation [22], was still secreted exclusively when the cells were pretreated with LPS, suggesting other factor(s) controlled by LPS were required for the response (Fig 1A). The secretion of both cytokines was inhibited by CP-456,773 [21]. Thus, both the requirement for LPS priming and sensitivity to CP-456,773 suggested that NLRP3 mediated the response to *C. difficile* in hMDM.

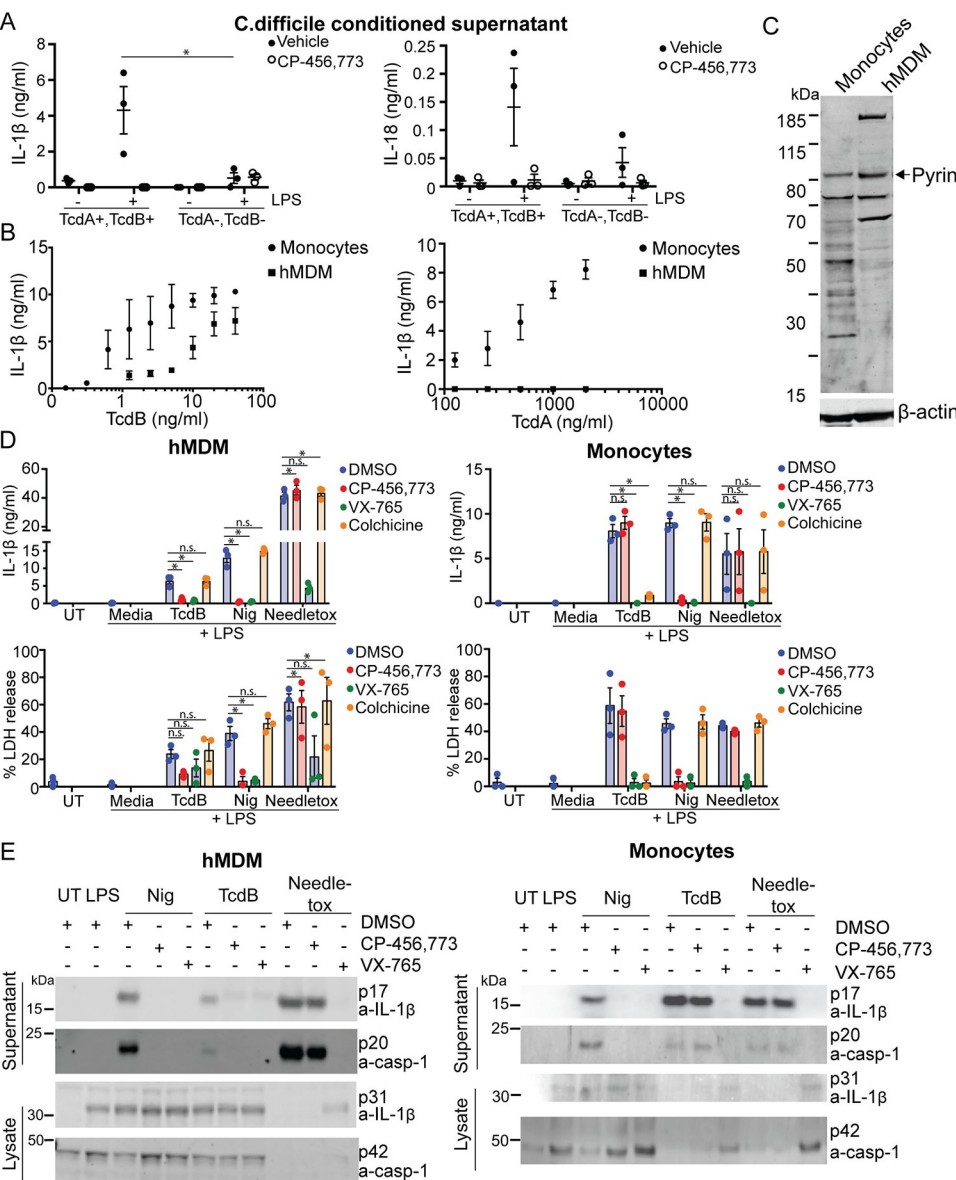

**Fig 1. TcdB, but not TcdA, triggers CP-456,773 sensitive IL-1β release in hMDM.** (**A**) hMDM incubated with conditioned supernatant from toxin-proficient (TcdA+,TcdB+) or toxin-deficient (TcdA−,TcdB−) *C. difficile* for 4 h, then the supernatant assessed for IL-1β and IL-18. (**B**) LPS-primed (10 ng/ml, 3 h) monocytes or hMDM were treated with a dose titration of either TcdB or TcdA for 3 h. Supernatants were harvested and assessed for IL-1β. (**C**) Immunoblot of lysate from LPS treated (10 ng/ml, 3 h) monocytes or hMDM probed for either Pyrin or actin. Representative of 3 independent experiments. (**D**) LPS-primed (10 ng/ml, 3 h) monocytes or hMDM were preincubated for 15 min with the vehicle alone, CP-456,773 (2.5 μM), VX-765 (40 μM), or colchicine (2.5 μM), then stimulated with TcdB (20 ng/ml), nigericin (8 μM), or needletox (25 ng/ml) for 2.5 h. The supernatant was assessed for IL-1β and LDH activity or (**E**) IL-1β and caspase-1 cleavage by immunoblot. Mean and SEM shown for 3 donors or immunoblots representative of 3 independent experiments. * $p < 0.05$, n.s. not significant. The underlying data can be found in the summary data file in the tab Fig 1A, 1B and 1D.

## *C. difficile* toxin B, but not toxin A, mediates inflammasome activation in hMDM

We observed that hMDM released IL-1β and IL-18 only in response to supernatant from the toxin-proficient bacteria. To determine whether one or both toxins could trigger an

inflammasome response in hMDM, we incubated the cells with either recombinant toxin A (TcdA) or toxin B (TcdB). As a control, we incubated monocytes with both toxins, as they have previously been shown to respond to both TcdA and TcdB in a Pyrin inflammasome-dependent manner. As assumed, monocytes released IL-1β in response to both TcdA and TcdB (Fig 1B). Surprisingly, and in contrast to the monocytes, we found that TcdB, but not TcdA, induced an inflammasome response in hMDM (Fig 1B). Notably, TcdB triggered IL-1β release from hMDM at concentrations as low as 1 ng/ml. To ensure that the lack of TcdA-mediated inflammasome activation was not due to a failure of toxin uptake, we assessed TcdA-mediated Rac modification. To do so, we used a previously described monoclonal antibody that no longer recognizes Rac when its epitope is modified by the toxin [23]. The antibody was unable to bind to Rac in both monocytes and hMDM treated with the active forms of TcdB and TcdA, but not with mutants lacking glucosyltransferase activity, indicating that the cells took up the toxin (S1A Fig).

Given the disparity in inflammasome response between monocytes and hMDM, we assessed whether Pyrin was differentially expressed in the 2 cell types. As the commercially obtained Pyrin antibody detected multiple protein bands in immunoblots of hMDM lysates, we first validated the Pyrin antibody by cross-absorbing it against an immunoblot with lysate from HEK cells transiently transfected with either Pyrin or a vector control. The cross-absorbed antibodies were then used to consecutively probe an immunoblot of hMDM lysates, first with the Pyrin cross-absorbed antibody, then with the antibody absorbed against lysate from vector alone transfected HEK cells (S1B Fig). We determined that the specific Pyrin band was just above 80 kDa, correlating with the predicted size of human Pyrin. We then assessed Pyrin expression in both monocytes and macrophages from a further 3 donors and found that Pyrin expression was surprisingly higher in hMDM than in monocytes, even though we found no Pyrin-dependent inflammasome response. (Figs 1C and S1C).

## *C. difficile* toxin B inflammasome activation is NLRP3 dependent in hMDM

To further investigate the TcdB-mediated inflammasome response, we preincubated LPS-primed hMDM or monocytes with inhibitors against NLRP3 (CP-456,773), Pyrin (colchicine), or caspase-1 (VX-765) and then treated the cells with TcdB. We used nigericin, a potassium ionophore that activates NLRP3, as well as the NLRC4 activator needletox as specificity controls. The latter contains the *Salmonella typhimurium* T3SS needle protein PrgI fused to the N-terminus of anthrax lethal factor, which is delivered to the cytosol by anthrax protective antigen (PA). Both TcdB-dependent IL-1β and lactate dehydrogenase (LDH) release, a measure of pyroptosis, were inhibited by CP-456,773 but not colchicine (Fig 1D). The decrease in LDH was not significant, though this could be explained by the relatively low amount of LDH released, suggesting that induction of pyroptosis by TcdB was not efficient. In contrast, and as demonstrated previously, the response to TcdB in monocytes was not inhibited by CP-456,773 but was inhibited by colchicine, indicating a dependence on Pyrin (Fig 1D). To ensure that the observed IL-1β release was accompanied by caspase-1 activation, we also assessed the cleavage of IL-1β and caspase-1. In agreement with the IL-1β and LDH release results, we found that TcdB-mediated IL-1β and caspase-1 cleavage were CP-456,773 sensitive in hMDM, but not in monocytes (Fig 1E). We also assessed the actin cytoskeleton after TcdB treatment by staining the cells with Phalloidin. As predicted, TcdB disrupted the actin cytoskeleton similarly in both cell types, as demonstrated by a redistribution of the Phalloidin signal. In LPS-treated cells, actin was distributed through the cytosol, while treatment with the toxin leads to accumulation of Phallodin puncta and redistribution to the edges of the cell (S1D Fig). In this study, we used

a shorter time period than has been used previously to differentiate the monocytes to hMDMs based on the protocol from Xue and colleagues [24]. To ensure that the observed effects on Pyrin activation were not due to this change, we also tested hMDM differentiated in M-CSF for 7 days rather than the 4 days used throughout this study and found that the response to both TcdA and TcdB in 7-day hMDM was the same as those generated in 4 days (S1E Fig).

## Monocytes, but not hMDM, release IL-1β in response to Pyrin inflammasome triggers

The lack of IL-1β release from hMDM in response to TcdA, coupled with the observation that TcdB activated the NLRP3 inflammasome instead, suggested that hMDM did not respond to Pyrin inflammasome triggers. To confirm this, we investigated whether TcdA or BAA-473 (11-oxo-12S-hydroxylithocholic acid methyl ester), a predicted microbe-derived bile acid metabolite and a recently identified Pyrin activator [25], could trigger an inflammasome response in hMDM. As expected, we observed CP-456,773-independent TcdA-mediated IL-1β release in monocytes (Fig 2A), but TcdA did not trigger IL-1β release in hMDM. Similarly, BAA473 stimulated IL-1β release in monocytes but not hMDM (Fig 2B). This was inhibited by VX-765 and colchicine but not CP-456,773 (Fig 2B), supporting the previous finding that the inflammasome response to BAA-473 is mediated by Pyrin, and demonstrating that hMDM do not mount a Pyrin inflammasome response.

## The inflammasome response to TcdB in BLaER1 cells and THP1 macrophages is dependent on NLRP3

Our results thus far demonstrated that NLRP3 is the responding inflammasome sensor to TcdB in hMDM but relied solely on compound-based inhibition. Therefore, we used

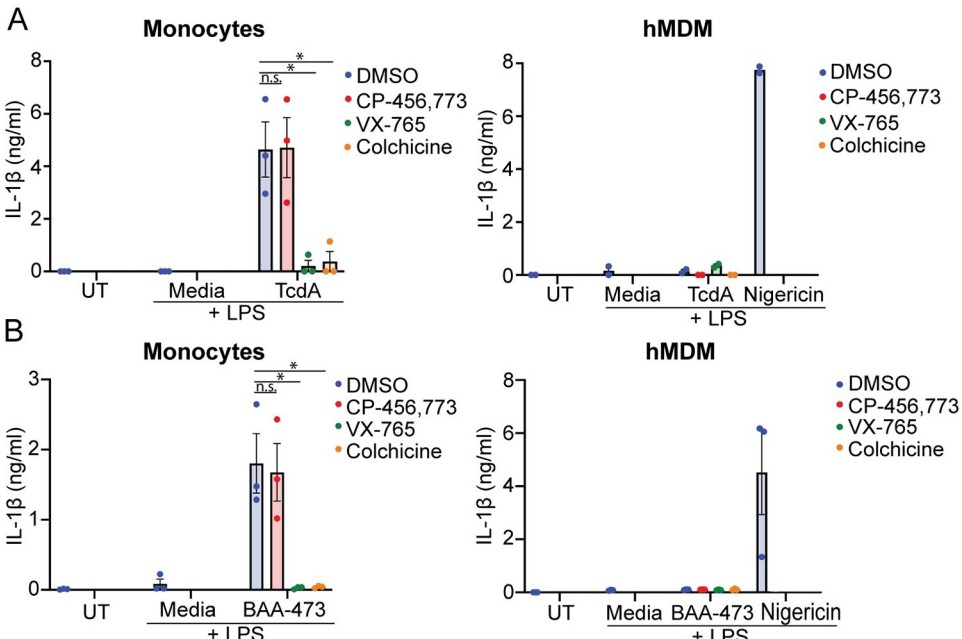

**Fig 2. TcdA and BAA-473 trigger IL-1β release in monocytes but not hMDM.** (**A**) IL-1β release from LPS-primed monocytes or hMDM stimulated with TcdA (500 ng/ml, 2h, both) or nigericin (hMDM only). (**B**) IL-1β release from LPS-primed monocytes or macrophages stimulated with the bile acid analogue BAA473 (10 μM, 2 h, both) or nigericin (hMDM only). Mean and SEM of 3 independent donors shown for monocyte experiments and hMDM stimulated with BAA473, mean from 2 independent donors shown for hMDM stimulated with TcdA, $*$ $p < 0.05$, n.s. not significant. The underlying data can be found in the summary data file in the tab Fig 2A and 2B.

genetically modified human macrophage cell lines to determine whether NLRP3 or Pyrin mediates the inflammasome response to TcdB. Initially, we sought a model cell line that had a robust inflammasome response to TcdB. We investigated whether the BLaER1 cell line, a recently established monocyte/macrophage cell line [26], would respond similarly to TcdB to the hMDM. We found that wild-type (WT) BLaER1 cells released IL-1β in response to treatment with TcdB, as well as activating caspase-1 as measured by a caspase-1 activity assay (Fig 3A). Similar to hMDM, TcdB-dependent IL-1β release and caspase-1 activity were inhibited by CP-456,773. These data demonstrate that TcdB activates an inflammasome response in BLaER1 similar to hMDM.

We next used the BLaER1 cell line to determine the propensity of TcdB to trigger ASC speck formation, another hallmark of inflammasome activation. Accordingly, we generated a BLaER1 cell line overexpressing ASC-mCherry and stimulated it with either TcdB or nigericin in the presence or absence of CP-456,773. This experiment was also performed in the presence of the caspase-1 inhibitor VX-765 to prevent cell death of inflammasome-activated cells. Following stimulation, the cells were fixed, and the number of ASC specks was quantified by microscopy, followed by normalization to the number of nuclei in each image. Consistent with our other data, we found that TcdB and nigericin also caused CP456, 773-sensitive ASC speck formation in these cells (Fig 3B).

To determine the sensor responsible for the TcdB-mediated inflammasome response, we used either NLRP3, caspase-4 double knockout (KO) or Pyrin KO BLaER1 cells. The NLRP3, caspase-4 double KO cells were reconstituted with either NLRP3, the inactive walker A/B NLRP3 mutant, or a vector alone control. NLRP3 expression was confirmed by immunoblot (Fig 3C). As done previously, we primed these cells with LPS, incubated them with either TcdB, nigericin, or needletox, and assessed IL-1β release. In agreement with our findings in hMDM, only the cells reconstituted with active NLRP3, but not the walker A/B mutant, were able to respond to TcdB and nigericin (Fig 3D). In contrast, all cell lines responded equally to the NLRC4 trigger needletox (Fig 3D) and secreted similar levels of TNFα in response to LPS (Fig 3D). We next ensured that the loss of response to TcdB was not due to the absence of caspase-4. Accordingly, we tested the inflammasome response to TcdB in the caspase-4 KO BLaER1 cells. In contrast to the NLRP3, caspase-4 double KO, these cells still released IL-1β, demonstrating that caspase-4 deficiency did not account for the loss of response to TcdB (S2A Fig). These results confirm that the TcdB-mediated inflammasome response was dependent on the expression of active NLRP3.

To confirm that Pyrin was not required for the TcdB-driven inflammasome response, we used lentiviral transduction to reconstitute Pyrin KO BLaER1 cells with either Pyrin fused in frame to a FLAG tag or with an empty vector control and confirmed Pyrin expression by immunoblot (Fig 3E). We found that the inflammasome response to TcdB was unaffected by the absence of Pyrin, indicating that Pyrin does not play a role in the inflammasome response to TcdB in these cells (Fig 3F). Similarly, Pyrin expression did not affect the inflammasome response to nigericin or needletox (Fig 3F).

BLaER1 cells can secrete IL-1β in response to TLR4 stimulation by LPS alone. To rule out that LPS contamination of the TcdB was responsible for the observed IL-1β release, we stimulated the cells with TcdB or nigericin following preincubation of the cells with TAK-242, a TLR4 inhibitor. TAK-242 did not block TcdB-mediated IL-1β release (S2B Fig), but TAK-242 completely abolished LPS-mediated TNFα secretion when applied before LPS stimulation (S2C Fig), demonstrating that NLRP3 mediated activation by TcdB is not due to LPS contamination or TLR4 activation.

Having established the requirement for NLRP3 in response to TcdB in the BLaER1 cell line, we sought to determine if this was true in another commonly used human macrophage cell

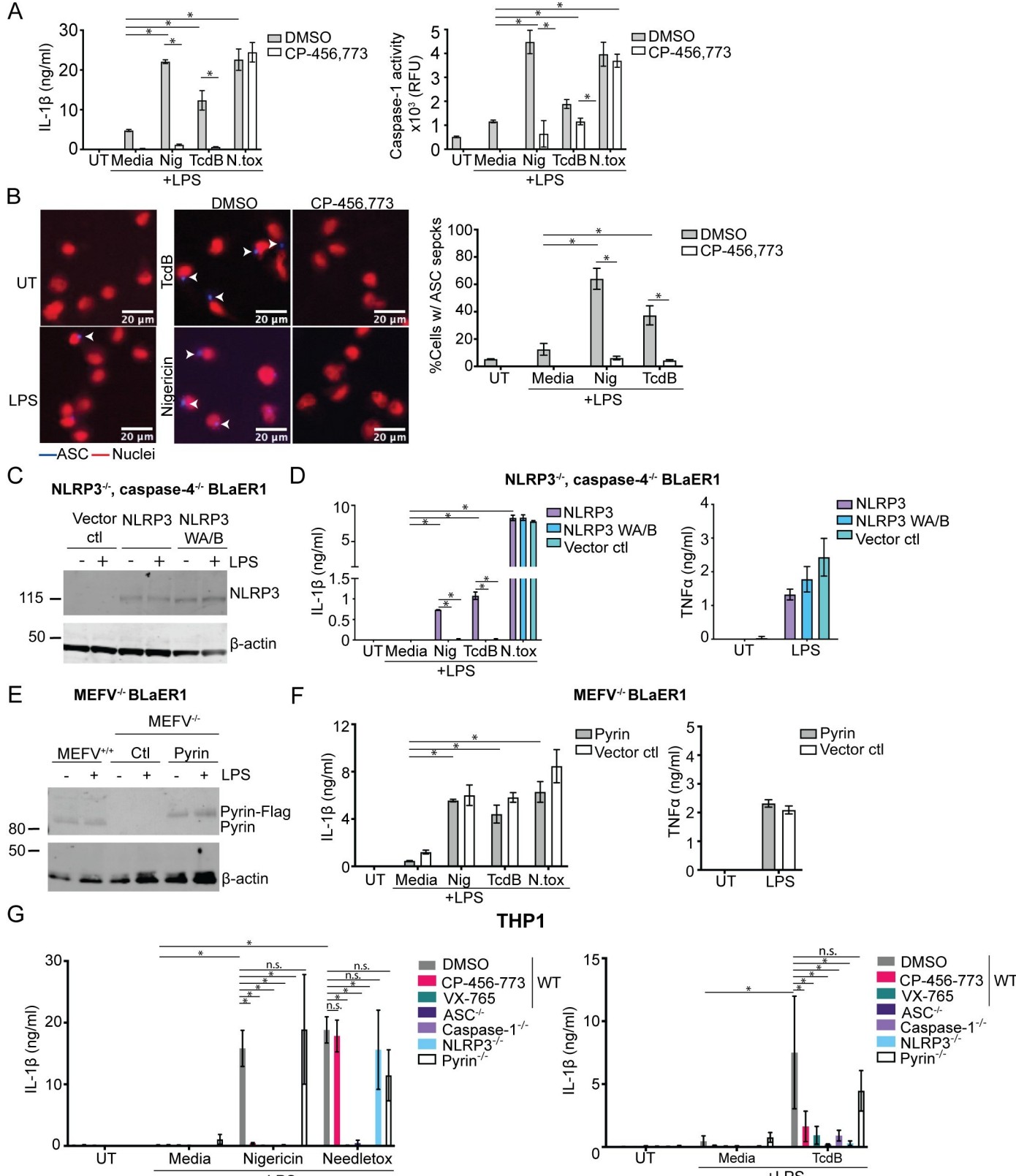

**Fig 3. NLRP3, not Pyrin, is the responding inflammasome sensor to TcdB in BLaER1 cells and THP1 macrophages.** (**A**) Differentiated WT BLaER1 cells were primed with LPS (100 ng/ml, 3 h), preincubated with either CP-456,773 (2.5 μM, 15 min), then activated with nigericin (8 μM), TcdB (20 ng/ml), or needletox (25

ng/ml each) for 2 h. IL-1β and caspase-1 activity were assessed from the harvested supernatants. (**B**) ASC-mCherry transduced WT BLaER1 cells treated as in (**A**). ASC is in blue; nuclei are red. Cells were then fixed and the number of ASC specks quantified by microscopy. (**C**) NLRP3 expression in differentiated BLaER1 cells (+/− 100 ng/ml LPS, 3 h) was assessed by immunoblot. (**D**) Differentiated caspase-4, NLRP3 double deficient BLaER1 cells reconstituted with either NLRP3-Flag, the NLRP3 walker A/B mutant (NLRP3 WAB-Flag), or the vector control treated as in (**A**) and the supernatants assessed for IL-1β or TNFα. Mean and SD of 3 technical replicates shown, representative of 3 independent experiments. (**E**) Immunoblot of Pyrin expression in differentiated BLaER1 cells (+/− 100 ng/ml LPS, 3 h). The Pyrin-deficient cells were reconstituted with Pyrin-Flag. Representative of 3 independent experiments. (**F**) Differentiated Pyrin-deficient BLaER1 cells reconstituted with either Pyrin-Flag or the vector control treated as in (**A**) and the supernatants assessed for IL-1β or TNFα. Mean and SD of 3 technical replicates shown, representative of 3 independent experiments. (**G**) LPS-primed WT THP-1s or the listed KOs were activated with inflammasome activators for either 1.5 h (nigericin, needletox) or for 8 h (TcdB). Supernatants were assessed for IL-1β. Where used, CP-456,773 (2.5 μM) and VX-765 (40 μM) were preincubated with the cells for 15 min prior to addition of the inflammasome activators. Mean and SEM of 3 independent experiments shown, * $p < 0.05$, n.s. not significant. The underlying data can be found in the summary data file in the tab Fig 3A, 3B, 3D, 3F and 3G.

line, THP-1. We first titrated TcdB on PMA differentiated THP-1 cells and found that it required both a higher concentration of TcdB (2 ug/ml) to trigger IL-1β release as well as a longer incubation time (8 h). Having established this, we tested several THP-1 CRISPR KO cell lines to determine the requirements for TcdB-mediated inflammasome activation. PMA-differentiated WT THP-1 cells or THP-1 cells deficient in ASC, caspase-1, NLRP3, or Pyrin were primed with LPS (200 ng/ml, 3 h), then incubated with TcdB, nigericin, and needletox. WT cells were also preincubated with CP-456,773 to determine if it had the same effect as ablation of NLRP3. Echoing our previous data, the TcdB-triggered inflammasome response in THP-1 cells was CP-456,773 sensitive and NLRP3 dependent (Fig 3G). As anticipated, this response also required ASC and caspase-1, but not Pyrin. Nigericin similarly was NLRP3 dependent, while the NLRC4 trigger only required ASC and caspase-1, demonstrating the effect of NLRP3 ablation was specific (Fig 3G). Furthermore, the different KO lines all secreted TNFα in response to LPS, indicating that the loss of response was not due to a lack of response to LPS (S2D Fig). Collectively, these results show an absolute requirement for NLRP3, but not Pyrin, in the TcdB-mediated inflammasome response in human THP-1 cells.

Having established that TcdB activates NLRP3, but not Pyrin, we next determined whether NLRP3 activation required the activity of the TcdB glucosyltransferase domain (GTD) against Rho, as found for Pyrin. Thus, we incubated LPS-primed hMDM with TcdB or a variant of TcdB containing inactivating mutations in the glucosyltransferase domain, D286N and D288N (TcdB NXN), which does not inactivate Rho, Rac, or Cdc42. We determined that both TcdB and TcdB NXN induced IL-1β release (S2E Fig). Interestingly, the NXN variant of TcdB appeared a more proficient NLRP3-inflammasome activator than the WT toxin, though the difference was not significant. In contrast, TNFα secretion was unchanged by either toxin compared to LPS (S2E Fig). This demonstrates that in contrast to Pyrin activation, GTD activity, and subsequent Rho inhibition are not required for the TcdB-mediated inflammasome response in hMDM.

## TcdA and TcdB elicit an NLRP3-independent inflammasome response in BMDM and peritoneal macrophages

The observation that neither TcdB nor TcdA could activate Pyrin in hMDM was unexpected. To test if this was specific to human cells, we next analyzed toxin-dependent inflammasome responses in differentiated macrophages from bone marrow (BMDM) as well as isolated peritoneal macrophages (PMs) from either WT or NLRP3 KO mice. We stimulated them with either TcdA or TcdB, using nigericin and poly (dA:dT) transfection as specificity controls for NLRP3 and AIM2, respectively.

TcdA and TcdB triggered IL-1β release in BMDM from both WT and NLRP3 KO cells, indicating that the response was NLRP3 independent (Fig 4A). This was also true in PMs, which showed little difference in toxin-mediated IL-1β release between WT and NLRP3 KO

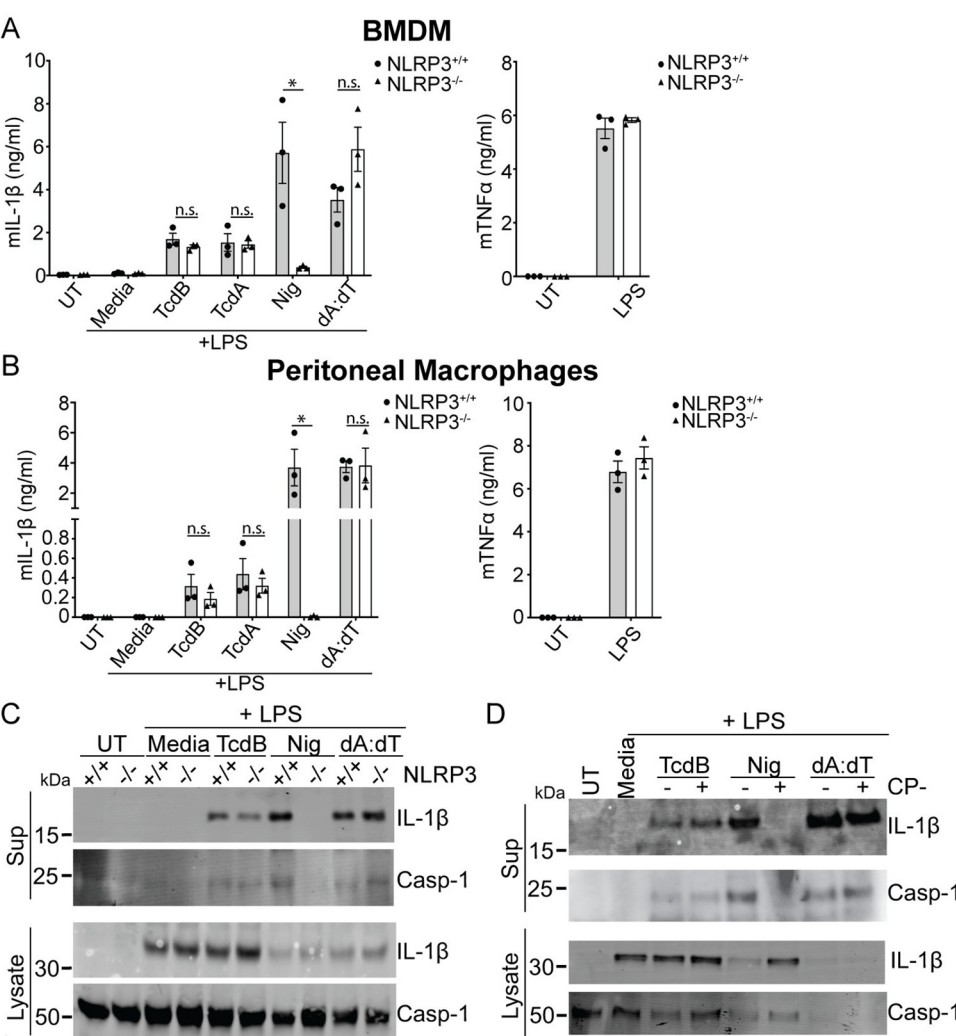

**Fig 4. TcdB triggers a NLRP3-independent inflammasome response in murine macrophages.** IL-1β release from WT and NLRP3-deficient BMDM (**A**) or PMs (**B**) primed with LPS (200 ng/ml, 3 h), then activated with nigericin and TcdB for 2 h or dA:dT for 4 h. (**C**) Caspase-1 and IL-1β immunoblots of precipitated supernatant or cell lysate from WT and NLRP3-deficient BMDM treated as in (**A**). (**D**) Caspase-1 and IL-1β immunoblots of precipitated supernatant or cell lysate from LPS primed WT BMDM either untreated or pretreated with CP-456,773 (2.5 μM, 30 min), then stimulated as in (**A**). Mean and SEM of 3 independent experiments shown, immunoblots are representative of 3 independent experiments. The underlying data can be found in the summary data file in the tab Fig 4A and 4B. BMDM, bone marrow–derived macrophage; LPS, lipopolysaccharide; PM, peritoneal macrophage; WT, wild-type.

cells (Fig 4B), while the inflammasome response to nigericin was ablated entirely in the NLRP3 KO. At the same time, there was no difference in IL-1β release in response to transfected dA:dT and no difference in TNFα secretion in response to LPS. We also assessed IL-1β and caspase-1 cleavage in WT and NLRP3 KO BMDM following LPS priming and stimulation with TcdB, nigericin, or dA:dT. Similarly, we found no differences between the 2 genotypes when stimulated with TcdB or the specificity control dA:dT, while nigericin-mediated caspase-1 cleavage was ablated in the NLRP3 KO (Fig 4C).

Given that we rely on CP-456,773 to determine the role of NLRP3 in response to TcdB in human primary macrophages, we also investigated whether, despite the lack of NLRP3 dependence, CP-456,773 could affect the response to TcdB in BMDM. Therefore, WT BMDM were

primed with LPS preincubated in the presence or absence of CP-456,773 and then incubated with TcdB, nigericin, and dA:dT. As with the NLRP3 KO cells, CP-456,773 did not affect TcdB- or dA:dT-mediated caspase-1 or IL-1 β cleavage but completely inhibited nigericin-mediated cleavage (Fig 4D). Therefore, the TcdB-mediated inflammasome response in murine macrophages is both NLRP3 independent and insensitive to CP-456,773. Collectively, these results show that, in our hands, the inflammasome response to both TcdA and TcdB in murine macrophages is independent of NLRP3. Given that the Pyrin inflammasome was demonstrated to be the responding sensor in other studies [3,20], the response we observe is likely dependent on Pyrin.

## Prolonged incubation with LPS or type I and II interferons increases Pyrin expression in hMDM

It was surprising that, in contrast to monocytes, neither TcdA, TcdB, or BAA-473 triggered a Pyrin inflammasome response in hMDM. Thus, we next investigated whether inflammatory conditions increased the expression of Pyrin and thus potentially enable its activation. Pro-inflammatory signaling molecules activating either the NF-κB or IRF transcription factors have been demonstrated to increase Pyrin expression in PBMCs [27]. To determine whether the activation of either of these pathways could increase Pyrin expression in hMDM, we treated the cells with LPS, Pam3CSK4, TNFα, IFN-β, and IFN-γ, as well as IL-4 and IL-10 for 5 or 18 h and assessed Pyrin expression by immunoblot. Notably, only LPS increased Pyrin expression after 5 h, while LPS, IFN-β, and IFN-γ increased Pyrin expression after 18 h (Fig 5A). In contrast, TNFα and Pam3CSK4 did not change Pyrin expression (Fig 5A). Thus, only stimuli that signal through IRF family transcription factors triggered an increase in Pyrin expression. To ensure that these molecules were functional in our system, we assessed the expression of IL-1β, a target of NF-κB, in response to LPS and Pam3CSK4. We determined that both were able to increase IL-1β expression (Fig 5A), demonstrating that they were functional but only LPS treatment increased Pyrin expression.

Previous studies had shown that the Pyrin promoter contains an ISRE element that can be activated by both TRIF and IFN signaling [27], suggesting that the increase in Pyrin expression observed can be due to increased transcription. We investigated this by stimulating the cells with either LPS or Pam3 for 12 h, then assessing mRNA copy number for Pyrin by qPCR, using IL-1β as a control. We observed that LPS, but not Pam3CSK4, caused an increase in *MEFV* transcripts compared to the untreated cells (Fig 5B). In contrast, both LPS and Pam3CSK4 increased IL-1β transcription, demonstrating that the increase in *MEFV* transcript was specific to LPS (Fig 5B). It is, therefore, likely that the increase in Pyrin expression is driven by increased gene transcription.

## LPS and interferons prime activation of the Pyrin inflammasome in hMDM

We tested whether increased Pyrin expression would be sufficient to enable Pyrin inflammasome activation. Accordingly, we primed hMDM with LPS for either 3 or 18 h, preincubated them with DMSO, CP-456,773, VX-765, or colchicine, and then treated them with TcdA, BAA-473, or nigericin. As we had observed previously, neither TcdA nor BAA-473 triggered an inflammasome response after 3 h of LPS priming (Fig 6A). In contrast, after 18 h, both TcdA and BAA-473 triggered robust release of IL-1β that was sensitive to colchicine and thus dependent on Pyrin (Fig 6A). By comparison, nigericin mediated CP-456,773-sensitive IL-1β release after both 3 and 18 h of LPS priming but was not affected by colchicine (Fig 6B). Similarly, pretreating hMDM with IFN-β for 18 h was sufficient to render TcdB-mediated IL-18 release insensitive to CP-456,773, in contrast to the response at 3 h. However, TcdB-mediated

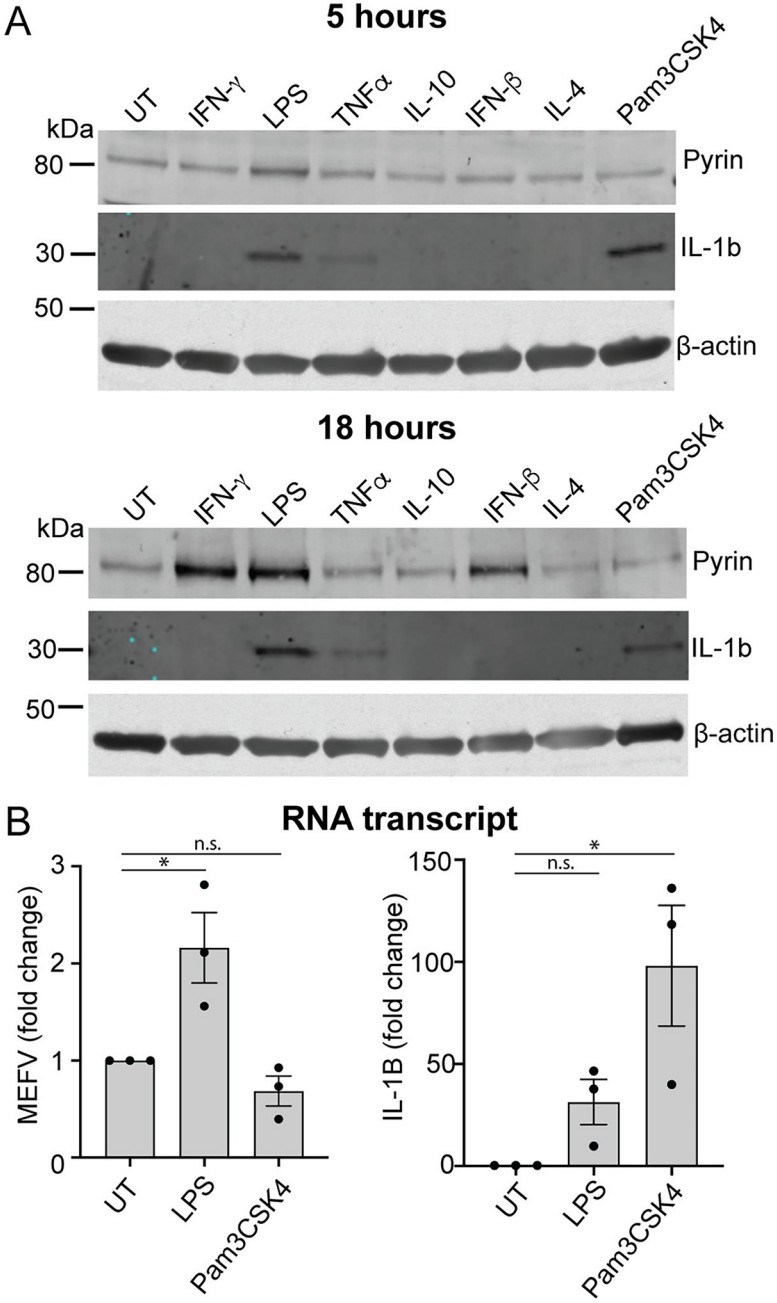

**Fig 5. LPS, type I and type II interferons increase Pyrin expression and enable Pyrin activation in human macrophages.** (**A**) Pyrin and IL-1β expression in hMDM treated with either IFN-γ (200 U/ml), LPS (10 ng/ml), TNFα (50 ng/ml), IL-10 (100 ng/ml), IFN-β (5,000 U/ml), IL-4 (1,000 U/ml), or Pam3CSK4 (20 ng/ml) for either 5 or 18 h. Representative of 3 independent experiments. (**B**) Pyrin (*MEFV*) or IL-1β (*IL1β*) transcript from hMDM-treated LPS (10 ng/ml) or Pam3CSK4 (20 ng/ml) for 12 h. Mean and SEM of the fold change of 3 experimental replicates shown. The underlying data can be found in the summary data file in the tab Fig 5B.

IL-18 release at both 3 h and 18 h was inhibited by colchicine. This is in contrast to what was seen with LPS and suggests that the NLRP3 inflammasome response can partially be inhibited by colchicine in the context of pretreatment with IFN- β. These results indicate that IFN- β could also prime a Pyrin response (S3A Fig).

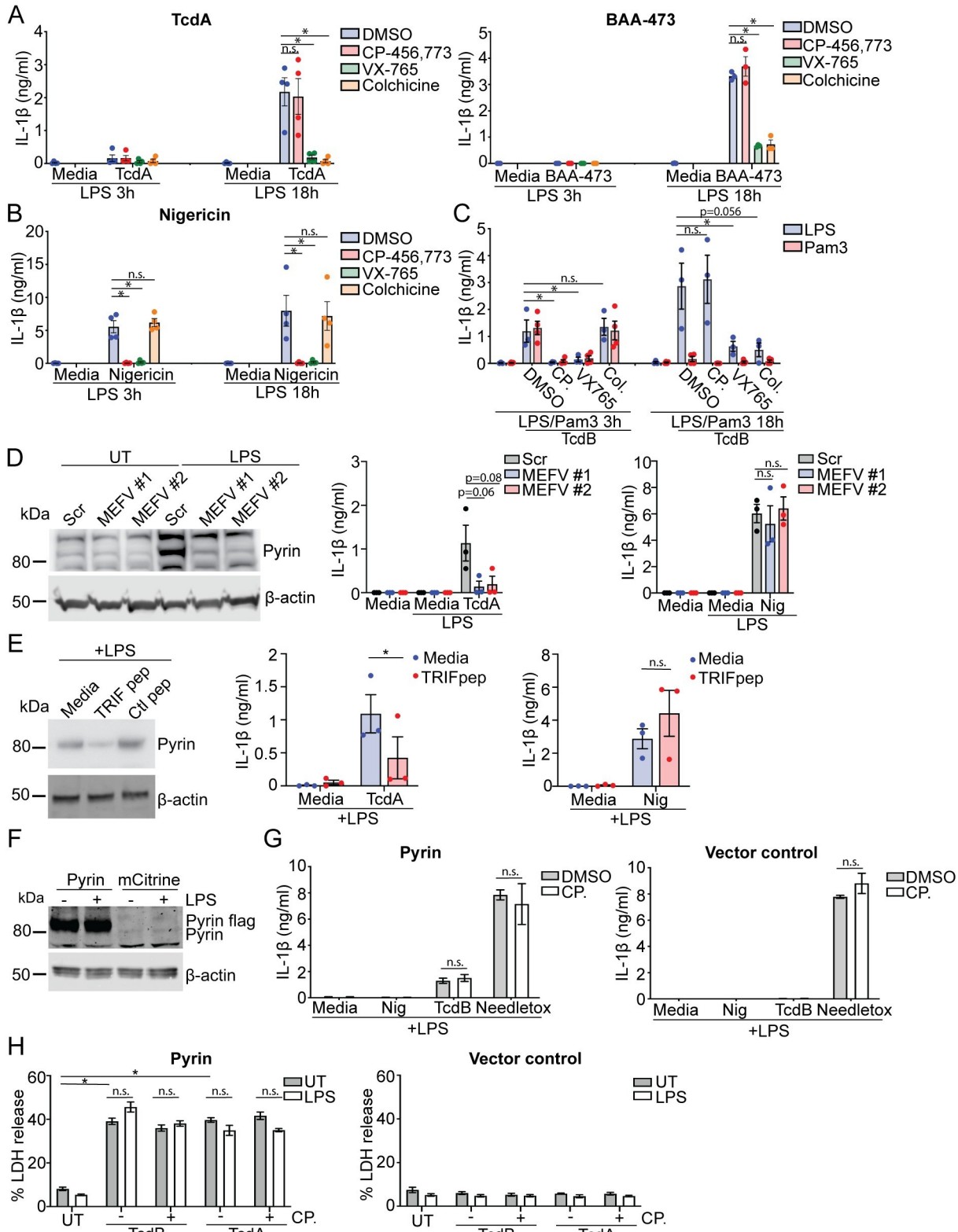

**Fig 6. Increased Pyrin expression is required for Pyrin activation in human macrophages.** (**A**) IL-1β release from LPS-primed (10 ng/ml, 3 or 18 h) hMDM were preincubated with compounds as noted previously, then stimulated with either TcdA (200 ng/ml), BAA-473 (10 μM), or (**B**) nigericin (8 μM) for 2.5 h. (**C**) IL-1β release from hMDM primed with either LPS (10 ng/ml) or Pam3Cys4K (20 ng/ml) for 3 h or 18 h and stimulated with TcdB (20 ng/ml). (**D**) hMDM were transfected with siRNA targeting Pyrin mRNA (MEFV#1 and MEFV#2) or the scrambled

control and assessed for Pyrin expression by immunoblot or stimulated a previously following primed with LPS (10 ng/ml) for 18 h. (**E**) Expression of Pyrin in hMDM incubated with pepinhTRIF or the scrambled control peptide (25 µg/ml, 5 h), then treated with LPS (10 ng/ml, 18 h). Representative of 3 donors. IL-1β release from cell pretreated with pepinhTRIF or the scrambled control peptide (25 µg/ml) in response to LPS priming (10 ng/ml, 18 h) followed by stimulation with either TcdA (200 ng/ml) or nigericin (8 µM) for 2.5 h. LPS-primed (100 ng/ml, 3 h) differentiated caspase-4, NLRP3 double-deficient BLaER1 cells reconstituted with either Pyrin or the vector control were (**F**) lysed and assessed for Pyrin expression by immunoblot or (**G**) incubated with nigericin (8 µM) TcdB (20 ng/ml) or needletox (25 ng/ml) for 2.5 h +/− CP-456,773 (2.5 µM, 15-min incubation), and the supernatants were assessed for IL-1β. (**H**) LDH release from Pyrin or vector alone reconstituted BLaER1 cells primed with LPS (3 h, 100 ng/ml) or left unprimed and stimulated with TcdB (20 ng/ml) or TcdA (1 µg/ml) for 2 h. For (**A**-**E**) and (**H**) mean and SEM of 3–4 experimental replicates shown. For (**G**) mean and SD of 3 technical replicates shown, representative of 3 independent experiments. * $p < 0.05$, n.s. not significant. The underlying data can be found in the summary data file in the tab Fig 6A–6E, 6G and 6H. LDH, lactate dehydrogenase; LPS, lipopolysaccharide; siRNA, small interfering RNA.

We next sought to determine whether restoration of the Pyrin inflammasome response was specific to a stimulus that increased Pyrin expression. We thus compared the ability of TcdB to activate Pyrin in hMDM primed with Pam3CSK4 to those primed with LPS. As done previously, hMDM were preincubated with different inhibitors to determine the responding inflammasome sensor. TcdB triggered CP-456,773-sensitive inflammasome activation after 3 h of priming with either LPS or Pam3CSK4 (Fig 6C). Comparative stimulation after 18 h of priming led to a Pyrin-dependent response in the LPS-primed cells, while the Pam3CSK4-primed cells failed to activate any inflammasome response (Fig 6C), demonstrating that increased Pyrin expression correlated with Pyrin reactivation. We then determined whether a decrease in NLRP3 expression contributed to the change of inflammasome response to TcdB but found that NLRP3 expression was comparable between 3 h and 18 h post-LPS treatment (S3B Fig).

## Increased Pyrin expression is necessary for the Pyrin response in hMDM

To determine whether the increase in Pyrin expression was a requirement for Pyrin activation, we transfected hMDM with 2 distinct small interfering RNAs (siRNAs) against Pyrin or a scrambled control 24 h before priming with LPS. Notably, both Pyrin-targeting siRNAs effectively prevented the LPS-mediated increase in Pyrin expression. Still, they did not reduce it further than the untreated control, while the scrambled control had no effect (Fig 6D). siRNA-transfected hMDM were primed with LPS for 18 h and then stimulated with TcdA or nigericin. We observed that the 2 Pyrin siRNAs, but not the scrambled control, prevented TcdA-mediated IL-1β release. In contrast, neither of the Pyrin-targeting siRNAs, nor the control siRNA, affected nigericin-mediated inflammasome activation (Fig 6D), establishing that decreasing Pyrin expression is sufficient to inhibit Pyrin activation specifically.

As Pyrin expression increased after treatment with LPS or interferons, but not Pam3CSK4 or TNFα, we hypothesized that the increase in LPS-dependent Pyrin expression likely required the TRIF signaling pathway. We tested our hypothesis by investigating whether blocking TLR4-mediated TRIF signaling prevents the LPS-dependent increase in Pyrin expression and the Pyrin inflammasome response. We pretreated hMDM with pepinhTRIF, a peptide that prevents the interaction of TRIF with its downstream interaction partners. We then incubated these cells with LPS for 18 h before assessing Pyrin expression and activation. We found that treatment with pepinhTRIF, but not a control peptide, reduced LPS-mediated Pyrin expression (Fig 6E). We then stimulated the cells with either TcdA or nigericin. We found that only the TcdA-mediated inflammasome response was inhibited by pretreatment with the pepinhTRIF, whereas the nigericin-mediated IL-1β release was unaffected (Fig 6E). These results demonstrate that the LPS-stimulated increase in Pyrin expression is TRIF mediated and that blocking this is sufficient to reduce Pyrin activation in these cells.

Given that increased Pyrin expression is necessary for Pyrin reactivation in hMDM, we next sought to determine whether Pyrin overexpression alone would enable a Pyrin

inflammasome response. For this experiment, we used the caspase-4, NLRP3 double KO BLaER1 cells, which otherwise do not mount an inflammasome response to TcdB. We overexpressed Pyrin-FLAG in the caspase-4, NLRP3 double KO BLaER1 cells using reconstitution with a vector alone as a control. The Pyrin-reconstituted cells expressed more Pyrin than the control cells, which surprisingly had relatively low Pyrin expression. We suggest that this is most likely due to the clonal nature of this CRISPR-generated cell line (Fig 6F). We primed these cells with LPS and stimulated them with TcdB, nigericin, or needletox. TcdB triggered an inflammasome response in the Pyrin-reconstituted cells, but not in cells transduced with the vector alone (Fig 6G). Notably, this was not inhibited by CP-456,773. In contrast, Pyrin reconstitution had no effect on either NLRP3 or NLRC4 activation (Fig 6G). We then assessed LDH release from these cells in response to TcdB or TcdA in the presence or absence of LPS to determine if LPS priming was also required for the Pyrin inflammasome response. However, there was no difference in Pyrin activation between these 2 conditions to either toxin (Fig 6H). LDH release required Pyrin, as the cells transduced with vector alone did not release LDH in response to either toxin (Fig 6H). These results differed from our earlier observations where TcdB triggered an NLRP3-dependent inflammasome response in Pyrin KO BLaER1 cells overexpressing Pyrin. However, this experiment was performed in NLRP3-sufficient BLaER1 cells, and Pyrin reconstitution in those cells was closer to baseline as compared the vast overexpression of Pyrin seen in Fig 6F. Furthermore, when NLRP3 is present, the NLRP3 inflammasome response to TcdB might predominate in BLaER1. These results demonstrate that an increase in Pyrin expression is sufficient to enable Pyrin inflammasome response, and regulation of Pyrin expression is the primary factor controlling Pyrin inflammasome activation in hMDM.

## TcdB-mediated dephosphorylation of Pyrin is unaffected in hMDM

Neither TcdA, TcdB, or BAA-473 could activate Pyrin in hMDM in the absence of prolonged incubation with LPS or interferon, suggesting that one or both of the 2 known inhibitory mechanisms controlling Pyrin, either dephosphorylation of the S208 and S242 residues, or the less well-characterized mechanism related to the B30.2 domain, was restricting its activation. To determine the involvement of phosphorylation in regulating Pyrin activation in hMDM, we tested whether TcdB mediated dephosphorylation of Pyrin at serine 242 after priming with LPS for 3 h or 18 h. Notably, we found that Pyrin was dephosphorylated after treatment with TcdB regardless of the length of LPS priming, demonstrating that this is unlikely to be the mechanism preventing Pyrin activation (Fig 7A).

## The B30.2 domain restricts human Pyrin activation

Given that we observed no difference in the phosphorylation state of Pyrin after TcdB treatment, we investigated the role of the B30.2 domain in restricting Pyrin activation in human macrophages. We reconstituted Pyrin-deficient THP1 cells with constructs encoding either doxycycline-inducible human Pyrin (hPyrin) or murine Pyrin (mPyrin), which lacks the B30.2 domain due to a frameshift mutation in the *mefv* gene. These were fused in frame with a FLAG-tag, T2A peptide, and mCherry to report expression. A vector control was used to ensure that the responses observed were specific to Pyrin expression. Following doxycycline treatment, the cell lines were stimulated with Pyrin activators TcdA or BAA-473, or the NLRP3 activator nigericin and assessed for inflammasome activation by LDH release. TcdB was not used in this experiment as these cells express NLRP3, and so it would be unclear which sensor was responsible for the TcdB-mediated inflammasome response. We first assessed expression following doxycycline treatment by flow cytometry and found that distribution and frequency of mCherry expression for the hPyrin and mPyrin reconstituted cells

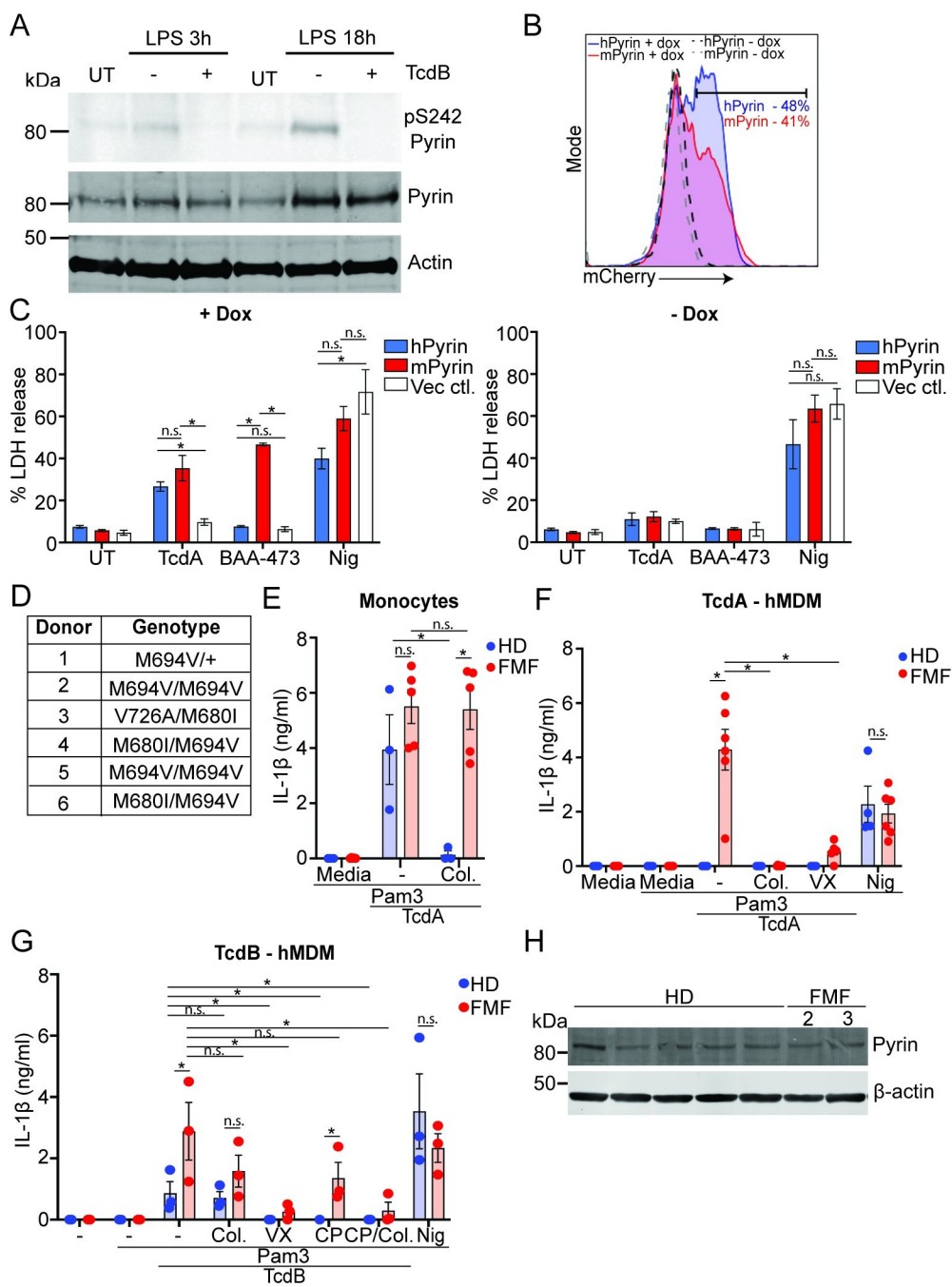

**Fig 7. The B30.2 domain regulates Pyrin activation in human macrophages and is disrupted by FMF mutations.**
(**A**) LPS-primed (10 ng/ml, 3 h or 18 h) hMDM were treated with TcdB (20 ng/ml, 1 h), then lysed and assessed for phosphorylation of Pyrin (S242), Pyrin, or actin by immunoblot. Representative of 3 independent experiments. (**B**) FACS profile of mCherry induction following doxycycline incubation, percentage positive cells shown. Representative of 2 independent experiments. (**C**) LDH release from Pyrin-deficient THP1 cells reconstituted with doxycycline-inducible hPyrin, mPyrin, or the vector control and treated with TcdA (1 μg/ml), BAA-473, (10 μM), or nigericin (8 μM) for 2 h. Mean and SEM from 3 independent experiments shown. (**D**) Table of the genotypes of the different FMF donors. (**E**) IL-1β release from monocytes from HDs or FMF donors (FMF, donors 2–6 used for the experiment) pretreated with Pam3CSK4 (25 ng/ml, 3 h), incubated with colchicine (2.5 μM) for 20 min, and stimulated with TcdA (1 μg/ml) for 3 h. IL-1β release from hMDM from HDs or FMF donors (FMF) pretreated with Pam3CSK4 (25 ng/ml, 3 h); incubated with CP-456,773 (2.5 μM), VX765 (40 μM), colchicine (2.5 μM), or CP-456,773 and colchicine together (TcdB only) for 20 min; and stimulated with (**F**) TcdA (1 μg/ml, FMF donors 1–6), (**G**) TcdB (20 ng/ml, FMF donors 3–6), or nigericin (8 μM) for 2 h. (**H**) Immunoblot for Pyrin expression in Pam3CSK4 (25 ng/ml, 3 h) primed hMDM

from HD or FMF (donors 2 and 3). For experiment in (**C**) mean and SEM from 3 independent experiments. For (**E-G**), mean and SEM shown for 3–6 independent donors. * $p < 0.05$, n.s. not significant. The underlying data can be found in the summary data file in the tab Fig 7C and 7E–7G. FMF, familial Mediterranean fever; HD, healthy donor; hPyrin, human Pyrin-flag; LDH, lactate dehydrogenase; LPS, lipopolysaccharide; mPyrin, mouse Pyrin-flag.

was similar (Fig 7B). Surprisingly, TcdA triggered similar levels of Pyrin-dependent cell death in both the hPyrin or mPyrin reconstituted cells, though there was a minor trend for increased activation in the cells expressing mPyrin (Fig 7C). More striking was the result using BAA-473, which activated mPyrin but not hPyrin (Fig 7C), suggesting that the B30.2 domain is more important to prevent activation by this ligand. Both responses were specific to expression of Pyrin, as dox-treated cells transduced with the vector control did not respond to either ligand, and there was no response in the absence of dox pretreatment (Fig 7C). All lines respond to nigericin to a similar level, though there was a minor trend showing decreased activation in the hPyrin compared to mPyrin cells (Fig 7C).

## FMF mutations enable activation of Pyrin in hMDM in the absence of LPS or interferon priming

The overexpression experiments in THP1 cells indicated that the B30.2 domain may be important in controlling Pyrin activation in hMDM. To investigate this in a more physiologically relevant setting, we tested the responsiveness of monocytes and hMDM derived from patients with FMF (Fig 7D) to Pyrin inflammasome activators compared to healthy donors. FMF is caused by gain-of-function mutations in the B30.2 domain of Pyrin, which could potentially disrupt Pyrin regulation in macrophages. We initially tested monocytes, as previous studies have demonstrated that while they respond to Pyrin stimuli similarly to healthy donors, the response cannot be inhibited by colchicine [16]. Consistent with these findings, TcdA triggered an inflammasome response in monocytes from both healthy donors and FMF patients, and the response was only colchicine dependent in healthy donors (Fig 7E). We next tested the response of hMDM to TcdB, TcdA, and nigericin. Notably, TcdA triggered a colchicine-sensitive inflammasome response in hMDM derived from FMF patients but not from healthy donors (Fig 7F), demonstrating that FMF causing mutations in the B30.2 domain disrupt the mechanism controlling Pyrin activation. Interestingly, the response to TcdB in the FMF donors was partially CP-456,773 dependent and partially colchicine dependent and was only completely blocked by VX-765, or a combination of CP-456,773 and colchicine (Fig 7G), suggesting that FMF mutations enable a Pyrin response in these cells, but NLRP3 is still the responding sensor in some cases. The response to nigericin was comparable between FMF and HD, demonstrating that the effect of the FMF mutations is restricted to the Pyin inflammasome (Fig 7F and 7G). We next assessed Pyrin expression to determine whether the increased Pyrin response in FMF donors was due to differences in Pyrin expression. However, there was no difference in Pyrin expression between the hMDM from FMF or HD (Fig 7H) for the donors tested. This experiment was unfortunately limited to only 2 donors due to the limited number of cells available from each donor. This suggests that the difference in responsiveness was not due to differences in Pyrin expression.

## Discussion

Pyrin responds to virulence factors that inhibit the RhoA signaling pathway. Gain-of-function mutants in the *MEFV* gene cause multiple genetic autoinflammatory disorders [12]. Understanding Pyrin regulation, particularly in the context of different human cell types, is therefore critical to understand how autoinflammation-associated mutations may disrupt these

mechanisms. In this study, we investigated the inflammasome response to the disease-causing *C. difficile* toxins TcdA and TcdB in hMDM. We determined that under steady state conditions, hMDM do not mount a Pyrin inflammasome response to either *C.difficile* toxin or the bile acid analogue BAA-473, suggesting that Pyrin cannot be activated in hMDM under steady state conditions generally. Pyrin activation could be established for all stimuli by prolonged stimulation with either LPS or type interferon, which licensed Pyrin activation by increasing Pyrin expression. Notably, hMDM derived from FMF patients with mutations in Pyrin responded to Pyrin triggers in the absence of LPS/interferon stimulation. These findings are in contrast to monocytes, which responded to both toxins in a Pyrin inflammasome–dependent manner, as has been shown previously [3].

The results of our study demonstrate that, alongside posttranslational mechanisms governing Pyrin activation, there is an additional requirement for transcriptional licensing specifically in hMDM. Transcriptional licensing required prolonged stimulation with LPS or interferons, but not other inflammatory stimuli including TNFα. This contrasts with a previous study in monocytes, where TNFα and Pam3CSK4 stimulation also increased Pyrin expression [27], as well as TNFα in mouse [28]. This data suggests that, in spite of the fact that the promoter region of the *MEFV* gene contains elements that are recognized by either interferon driven transcription factors or by NF-kB [27], increased Pyrin expression in hMDM was specific to activation of the TRIF/interferon pathway. Pyrin licensing in hMDM required an increase in Pyrin expression, as preventing the increase in Pyrin expression after stimulation with LPS inhibited Pyrin activation, while overexpression was sufficient to enable a Pyrin response. This is consistent with other inflammasomes, as activation of both NLRP3 and AIM2 is at least in part regulated by their expression [29,30]. However, it is unclear whether the increase in Pyrin expression is sufficient to enable its activation. Overexpression of pyrin in THP1 macrophages or BLaER1 cells was required for a Pyrin-dependent inflammasome response, even though these cells already expressed Pyrin. It is possible that increased expression alone enables Pyrin activation by enabling it overcome the restriction(s) limiting its activation, as these may have a finite capacity to restrict Pyrin activation. Conversely, LPS/ interferon licensing may provide additional signals such as alterations in Pyrin interactors or posttranslational modifications.

The lack of Pyrin activation observed in hMDM under steady state conditions suggested that posttranslational mechanisms were preventing its activation. There are 2 regulatory mechanisms restricting Pyrin activation identified so far: phosphorylation of S208 and S242 residues [10] or proposed regulation by the B30.2 domain [15]. Our data demonstrate that the B30.2 domain restricted Pyrin activation in hMDM. By comparing activation of hPyrin with mPyrin, which lacks the B30.2 domain, we determined that mPyrin was more easily activated by TcdA and BAA-473 even under conditions of overexpression. Surprisingly, the difference in activation between hPyrin and mPyrin with TcdA was minor. In comparison, BAA-473 strongly activated mPyrin but did not activate hPyrin. It is unclear why this result differs from that from the primary hMDM. However, the mechanism through which BAA-473 activates Pyrin is still unknown, and so could differ between hPyrin and mPyrin. Further research into the mechanism of Pyrin activation by BAA-473 may provide new data that explain this result. This suggests a differential regulatory role for the B30.2 domain between these stimuli.

FMF-causing mutations in the B30.2 domain also enabled Pyrin activation by TcdA and TcdB in hMDM. Importantly, using patient-derived primary cells, we could test these requirements at endogenous Pyrin levels, demonstrating that disruption of the B30.2 domain was enough to enable Pyrin activation without any requirement for priming or overexpression. Consistent with this, TcdB still triggered dephosphorylation of the S242 residue, demonstrating that this is not the mechanism restricting Pyrin activation in hMDM. How the B30.2

domain is differentially regulated between monocytes and macrophages is still unclear, as is how increased Pyrin expression in hMDM overcomes the B30.2 domain-mediated regulation. One possibility is the protein proline serine threonine phosphatase-interacting protein 1 (PSTPIP1), which binds to Pyrin and facilitates its oligomerisation and inflammasome formation [31]. It is possible that the expression level or availability of PSTPIP1 differs between these 2 cell types. Comparisons of the possible interaction partners or posttranslational modifications of Pyrin in monocytes and macrophages could provide further information on the function of the B30.2 domain controlling Pyrin activation.

The requirement for the B30.2 domain to restrict Pyrin activation in hMDM highlights a potentially important finding for the pathogenesis of FMF. FMF is characterized by periodic inflammatory flares resulting in joint swelling, skin lesions, and peritonitis among other symptoms [14]. Studies in monocytes from FMF patients show increased IL-1β secretion compared to healthy donors in response to TcdB and respond to lower doses of TcdB than healthy donors, indicating that the FMF mutations render Pyrin more sensitive to activation [32]. Furthermore, monocytes from FMF patients secrete more IL-1β in response to LPS stimulation alone (though this was NLRP3 dependent) [33], and some studies have suggested that Pyrin containing FMF mutations cause macrophages to be hyperinflammatory compared to those from healthy donors [34,35]. Here we identify a context, in macrophages, where FMF mutations in Pyrin enable an inflammasome response where WT Pyrin does not respond at all. Our data demonstrate that FMF-causing mutations enable Pyrin activation in hMDM in the absence of priming and the subsequent increase in Pyrin expression, which differs from Pyrin activation in hMDM from HD. This data is also consistent with initial findings from Shiba and colleagues, who showed that TcdA stimulated IL-1β release in hMDM from FMF patients but not from healthy donors [35]. Of note, this aberrant Pyrin activation is still inhibited by colchicine, which, consistent with previous findings [16], is not the case in monocytes. Given that FMF presents with tissue-based inflammation and occurs periodically, it is likely that it is Pyrin dysregulation in macrophages that underpins the initial inflammatory response. In support of this observation, colchicine, the first-line treatment for FMF, still blocks Pyrin activation in macrophages but not monocytes, providing an explanation for how it prevents FMF in the absence of Pyrin inhibition in monocytes. Research focusing on the role of macrophages in FMF will provide further evidence of their role in this disease.

A surprising finding from our study was that, in the absence of Pyrin activation, TcdB instead activated NLRP3 in these cells, demonstrating redundancy in the inflammasome system to detect this toxin. TcdB similarly triggered NLRP3 activation rather than Pyrin in both human macrophage cell lines we tested, BLaER1 and THP1, rather than Pyrin. Notably, unlike Pyrin activation, NLRP3 activation by TcdB did not require the enzymatic activity of the toxin, demonstrating that it activates Pyrin and NLRP3 through 2 distinct pathways. It was interesting to note that following prolonged priming, the response to TcdB became largely NLRP3 independent as well as Pyrin dependent. It is unclear why NLRP3 does not respond, as it was still expressed and activated in response to nigericin. One possibility is that prolonged priming may specifically inhibit activation of NLRP3 by the pathway triggered by TcdB. This demonstrates a cell type–specific divergence in the inflammasome response to TcdB.

The difference in Pyrin activation between monocytes and macrophages demonstrate another point of divergence in inflammasome activation between these cell types. They also differ in their responses to LPS, which activates NLRP3 in monocytes without a need for a second stimulus [26]. Monocytes are migratory cells that rely on actin rearrangement to reach sites of infection, where they contribute to the immune response and clearing the pathogen. It may thus be particularly relevant to monitor the functionality of the actin cytoskeleton with Pyrin, as inhibition of migration represents a disruption of a basic function of these cells. This

is consistent with Pyrin activation in neutrophils, another migratory immune cell type. Conversely, our results demonstrate that Pyrin is unable to respond in hMDM, even though inactivation of Rho will also impact the immune response and viral clearance. This suggests a more nuanced control of Pyrin activation in hMDM, which may limit Pyrin-driven autoinflammation, ensuring that Pyrin can only be activated after exposure to LPS or interferons. This could potentially occur through potential bile acid metabolites generated by bacteria [25], though further research will be required to determine if these are generated in vivo. How widespread this mechanism is will be elucidated by research focusing on Pyrin regulation in other cell types. Notably, the regulation of Pyrin observed in human macrophages was not evident in murine macrophages, which, similar to human monocytes, responded to TcdA and TcdB in a Pyrin-dependent manner. However, murine Pyrin lacks the B30.2 domain, and so may have lost the regulatory mechanism preventing Pyrin activation in hMDM. This demonstrates a further divergence in inflammasome responses between the 2 species, in addition to NLRC4 activation and NLRP3 responses [36,37].

Given that Pyrin does not seem to have an inflammasome-forming function in hMDM, it is unclear why Pyrin is nonetheless expressed. One possible explanation for this is that Pyrin has additional roles in the cell aside from forming an inflammasome. It has been suggested previously that Pyrin operates as a specialized adapter for autophagic machinery [38]. In this capacity, Pyrin associates with autophagic adapters ULK1 and Beclin1 to target substrates such as NLRP3 and capase-1 for autophagic degradation. Further investigation would be required to understand how M-CSF-driven Pyrin expression controls this phenomenon.

Our findings have implications for the pathogenesis of *C. difficile* infection. In the absence of prior priming, TcdB triggered either a Pyrin- or NLRP3-dependent inflammasome response depending on the cell type. This redundancy in the detection of TcdB is quite intriguing and suggests that inflammasome-mediated detection of TcdB is important in the response to the bacteria. Given that different strains of *C.difficile* express only TcdA, only TcdB, or both [39], it is also possible that disease severity alters depending on whether the bacteria express TcdB. Early NLRP3 activation in the macrophages prior to systemic penetration by the toxin may dictate the speed of the immune response or conversely enhance tissue damage. Furthermore, the detection mechanism for each sensor has different requirements, as Pyrin detects the activity of the glycosyltransferase domain through RhoA inactivation, while NLRP3-mediated TcdB detection is independent of glycosyltransferase activity. Our data demonstrate that Pyrin activation is differentially regulated in human and mouse, and so the inflammasome response to this infection in humans may differ from what has been shown in mouse models. Further studies in a model expressing a Pyrin homolog more closely resembling human Pyrin, such as the pig, are needed to determine the role of inflammasomes in *C.difficile* infection. Such studies will determine whether the inflammasome inhibitors currently being developed represent new treatments to prevent *C. difficile*-associated pathology or whether they pose an increased risk of *C. difficile* infection.

## Materials and methods

### Ethics statement

Ethics for the use of human material was obtained according to protocols accepted by the institutional review board at the University Clinic Bonn; local ethics votes Lfd. Nr. 075/14. No consent was taken as all donors were anonymous. Ethics for the use of human material from FMF patients was obtained from the Ethikkommission at the Charite Hospital Berlin; number EA1/007/17 to Dr. Karoline Krause. All donors gave informed, written consent.

## Reagents

LPS (Eb-ultrapure, 0111:B4), Pam3CSK4, pepinhTRIF, and TAK-242 were obtained from Invivo-Gen, nigericin was obtained from Invitrogen, and *Bacillus anthracis* PA was obtained from List Biological Laboratories. Colchicine was obtained from Sigma, and VX-765 was obtained from Sellekchem. BAA-473 was a gift from Dr. Canham (Novartis). DRAQ5 was purchased from eBioscience. TNFα, IFN-β, IFN-γ, M-CSF, IL-3, IL-4, and IL-10 were purchased from Immuno-tools. TcdA and TcdB from *C. difficile* strain VPI10463 were recombinantly produced and supplied by Prof. Ralf Gerhard [40]. Both toxins are identical to TcdA and TcdB from strain cdi630, which was used for infection assay. The HTRF kits for human IL-1β and TNFα were obtained from Cisbio; the ELISA kit for human and mouse IL-1β was obtained from R&D Systems. Both were used according to the manufacturer's instructions. For the *C. difficile* supernatant transfer, the following reagents were used: Butyric acid (Sigma-Aldrich: W222119-1KG-K), Various amino acids (Roth or Sigma), Iron sulfate heptahydrate (Sigma-Aldrich: 215422-250G), Triton-X 100 (Roth: 3051.2), M-Per (Sigma-Aldrich: 78501), Protease inhibitor (Sigma-Aldrich: 11836170001).

## Cells lines and tissue culture

The BLaER1 cells and THP-1 cells were maintained in complete RPMI (RPMI containing 10% heat-inactivated fetal calf serum, 1% Pen/Strep and 1% GlutaMAX, all obtained from Thermo Fisher). To trans-differentiate the BLaER1 cells to a macrophage-like cells, they were resuspended at $1 \times 10^6$ cells/ml in complete RPMI with 10 ng/ml IL-3, 25 U/ml M-CSF, and 100 nM β-Estradiol, and then 100 μl was plated in poly-L-lysine (Sigma, P8920) coated 96-well plates and incubated for 6 days to differentiate the cells. On day 6, the cells were used for experiments. The BLaER1 CRISPR KO cell lines were obtained from the laboratory of Prof. Veit Hornung and were generated as described in [26]. THP-1 cells were differentiated in full medium containing 50 ng/mL PMA. The THP-1 CRISPR KO cells were generated using trans-duction with lentiviruses based on pLenti CRISPR v2 [41] using sgRNAs: Caspase- 1-TACCA TGAGACATGAACACC, ASC–GCTGGATGCTCTGTACGGGA, NLRP3—CAATCTGAAGAAG CTCTGGT and Pyrin–TCTGCTGGTCACCTACTATG, followed by selection in 0.75 μg/mL puromycin. Monoclonal cell lines were generated by limited dilution and verified by Sanger sequencing and immunoblot. Doxycycline was used at 0.5 μg/ml and added to the cells for 5 h on the day prior to the experiment, then removed and the cells washed. The experiment was performed approximately 14 h later.

## Primary cell isolation and differentiation

Monocytes were purified from buffy coat preparations from healthy donors or from whole blood from either healthy donors or FMF patients. All donors were anonymous. The blood was mixed in a 2:3 ratio with PBS, layered onto a ficoll gradient, and centrifuged at 700*g* for 20 min without brake. The PBMC layer was extracted from the interface. After washing, it was incubated with CD14 conjugated magnetic beads (Milltenyi Bioscience) and purified using MACS columns (Milltenyi Bioscience) as per the manufacturer's protocol. The cells were then counted and resuspended for direct use or cultured for 3 days in RPMI containing 50 U/ml M-CSF at $2 \times 10^6$ cells/ml to generate hMDM. After the 3-day differentiation, hMDM were harvested and plated for experiments, then left to adhere overnight in RPMI containing 25 U/ml M-CSF.

## Inflammasome stimulation assays

**Primary monocytes/hMDM.** Cells were harvested and seeded the day before the assay. Before the experiment, the media was removed, and fresh media with or without a TLR

stimulus was added (LPS 10 ng/ml, Pam3CSK425 ng/ml) and incubated for 3 h. Next, compounds were added and incubated for 15 min (CP-456,773 2.5 uM, VX-765 40 uM, colchicine 2.5 μM). Inflammasome activators were subsequently added and incubated for 2.5h (TcdB 20 ng/ml, TcdA 1 μg/ml, BAA-473 10 μM, nigericin 8 μM, needletox 25 ng/ml, respectively). Plates were centrifuged at 450$g$ for 5 min, then the supernatant harvested for a cytokine or immunoblot analysis, and the cells lysed in RIPA buffer where applicable.

**BLaER1 cells.** Differentiated BLaER1 cells were seeded in 96-well plates precoated with poly-L-lysine. Before the experiment, the media was removed and fresh media with or without LPS (100 ng/ml) was added and incubated for 3 h. Next, compounds were added and incubated for 15 min (CP-456,773 2.5 μM, VX-765 40 μM, colchicine 2.5 μM). Inflammasome activators were subsequently added and incubated for 2.5 h (TcdB 20 ng/ml, nigericin 8 μM, PrgI/ PA 25 ng/ml, respectively). Plates were centrifuged at 450$g$ for 5 min, and then the supernatant was harvested for cytokine analysis.

THP1s were seeded in 24-well plates in the presence of 50 ng/mL PMA ($2 \cdot \times 10^5$ per well). Medium was replaced after 16 h; 24 h after this, cells were primed with 100 ng/mL LPS for 3 h, followed by treatment with 8 μM nigericin (NLRP3) or 100 ng/mL PA + 200 ng/mL LFn-PrgI (needletox, NLRC4) for 1.5 h, or 2 μg/mL TcdB for 8 h. Supernatants were cleared by centrifugation at 4˚ C, 1,000$g$ for 10 min and IL-1β and TNFα levels were quantified by ELISA. Where indicated, cells were treated with 2.5 μM CP-456,773 (CRID3, MCC950), or 40 μM Vx-765 for 30 min before and during stimulation. For the hPyrin/mPyrin comparison experiments, the cells were seeded following doxycycline stimulation and used the next day. These cells were stimulated directly with inflammasome stimulators without prior priming and supernatants harvested 2.5 h later.

## Cloning and molecular biology

The coding sequences for NLRP3, NLRP3 WA/B mutant and human Pyrin were all cloned into the pR vector in frame with a flag-tag, T2A peptide, and mCitrine. For doxycycline-inducible expression, the coding sequences for human or murine Pyrin were cloned into a TetO3 vector in frame with a flag-tag, T2A peptide, and mCherry and included a Blasticidin selection cassette. All vectors were produced by transforming DH5α *Escherichia coli*, selection of transformed clones using ampicillin, then purified using either Purelink Hipure Plasmid miniprep kit for small-scale production or Purelink Hipure plasmid maxiprep kit for high yield purification. All vectors were sequenced prior to use to ensure correct insertion and sequence.

## Sample preparation and immunoblotting

The supernatant from primary human monocytes, hMDM, or BMDMs ($2 \times 10^6$ cells/well in a 6-well plate) was harvested following inflammasome stimulation and the cells lysed in RIPA buffer (20 mM Tris–HCl (pH 7.4), 150 mM NaCl, 1 mM EDTA, 1% Triton X-100, 0.1% SDS, 0.5% deoxycholate, cOmplete protease and PhosSTOP (Roche) inhibitor). First, DNA was disrupted by sonication, then the lysate equivalent to $2 \times 10^5$ cells was mixed at a 1:4 ratio with 4× LDS buffer containing 10% sample reducing agent (Invitrogen). Samples were heated at 95˚C for 5 min and collected by centrifugation before loading.

Protein from supernatants were then precipitated by adding an equal volume of methanol and 0.25 volumes of chloroform, centrifuged for 3 min at 13,000$g$. Next, the upper phase was discarded, the same volume of methanol from the previous step was added, and the sample was centrifuged for 3 min at 13,000$g$. The pellet was then dried and resuspended in 1× LDS-sample buffer containing a 10% sample reducing agent (Invitrogen). Samples were heated at 95˚C for 5 min and collected by centrifugation before loading.

Proteins were separated by 4% to 12% SDS-PAGE in precast gels (Novex; Invitrogen) with MOPS buffer for proteins above 50 kDa or MES buffer for proteins below 50 kDa (Novex; Invitrogen). Proteins were transferred onto Immobilon-FL PVDF membranes (Millipore), and nonspecific binding was blocked with 3% BSA in Tris-buffered saline for 1 h, followed by overnight incubation with specific primary antibodies in 3% BSA in Tris-buffered saline with 0.1% Tween-20. For the phospho-Pyrin immunoblots, the transferred membranes were instead blocked in Tris-buffered saline containing 1% milk powder.

Primary antibodies were used as follows: NLRP3 (1:5,000; Cryo-2), human caspase-1 for lysate analysis (1:1,000; Bally-1), murine caspase-1 (1:1,000; casper-1) from Adipogen, Pyrin (1:1,000, *MEFV* polyclonal 24280-1-AP) from Proteintech, phospho-Pyrin S241 (1:500; ab200420) from Abcam, human IL-1β (1:1,000, AF-201-NA), and murine IL-1β (1:1,000, AF-401-NA) from R&D Bioscience, Rac (1:1,000, clone 102) from BD Transduction Laboratories, human caspase-1 for supernatant analysis (1:1,000, D57A2), Rac1/2/3 (1:1,000, rabbit poly-clonal #2465) from CST, actin (mouse or rabbit, both 1:1,000 dilution) from LI-COR Biosciences. Membranes were then washed and incubated with the appropriate secondary antibodies (IRDye 800CW, IRDye 680RD or HRP; 1:25,000 dilution; LI-COR Biosciences). In the case of caspase-1 detection or Pyrin detection, the membranes were incubated with washed and analyzed with an Odyssey CLx imaging system (LI-COR Biosciences) and ImageStudio 3 Software (LI-COR Biosciences). For the phospho-Pyrin immunoblots, the membrane was developed using western lighting plus-ECL and analyzed on a VersaDoc (Biorad).

## Cytokine measurements

Cytokines were measured either by ELISA or by HTRF as per the manufacturer's instructions. The kits used for ELISA were the human IL-1β (DY201), human TNFα (DY210), mouse IL-1β (DY401), or mouse TNFα (DY410), all from RnD Biosystems. HTRF kits used were human IL-1β (62IL1PEC) or human TNFα (62TNFPEB). All assays were read using a SpectraMAX i3 (molecular devices) using the additional HTRF cartridge. Human IL-18 measurements were performed using a cytokine bead array that was generated in our laboratory. The xMAP antibody coupling kit from Luminex (40–50016) was used to conjugate the capture IL-18 antibody (D0044-3, MBL) to the beads. IL-18 was then measured following the standard Luminex protocol and measured on a Magpix multiplexing unit (Luminex).

## siRNA transfection

hMDM were harvested by centrifugation (350*g*, 5 min) and washed twice in PBS. The cells were then aliquoted to have $1.2 \times 10^6$ cells per reaction and centrifuged again for 2 min at 3,000 rpm. The supernatant was discarded, and the cell pellet resuspended in 10.5 μL Buffer R with siRNA at 10 nM. Approximately 10 μL of reaction mix was loaded into the neon electroporator, and the pipette plugged into place within the electroporation tube containing 3 ml Buffer E. The electroporation set-tings were as follows: 1,400 V, 20 ms, and 2 pulses. Subsequently, the cells were transferred into 2 mL prewarmed antibiotic-free RPMI. After counting, the appropriate number of cells was seeded in 12-well or 96-well tissue culture plates and incubated for 24 h before experiments.

## Immunofluorescence and microscopy

Following treatment, the cells were washed once in PBS, then fixed in 2% PFA at 4°C overnight (for BLaER1 ASC speck analysis) or 4% PFA on ice for 30 min. The cells were then washed twice in PBS containing 20 mM Glycine, then twice in PBS. To stain for intracellular targets, the cells were permeabilized in 0.1% Triton X-100 for 5 min and blocked in intracellular staining solution (PBS +10% goat serum, 1% HI-FBS, and 0.1% Triton X-100) for 30 min RT. Next, we used Alexa-

647-conjugated Phalloidin (Invitrogen, A22287) for 30 min RT in an intracellular staining solution to stain actin. The cells were then washed (3× 5 min) and incubated with DAPI (1 μg/ml, 10 min) before being washed and imaged. For ASC speck detection, the fixed cells were incubated with DRAQ5 (eBioscience, 65-0880-96) for 5 min (1:2,000 dilution), then imaged directly. All imaging was performed with an Observer.Z1 epifluorescence microscope, 20× objective (dry, PlanApo-chromat, NA 0.8; ZEISS), Axiocam 506 mono, and ZEN Blue software (ZEISS). Image analysis of all ASC speck experiments was done using a cell profiler pipeline optimized to detect either ASC specks or nuclei. A minimum of 6 images was analyzed for each condition in each experiment.

## Retroviral transduction and fluorescent activated cell sorting

To produce the virus-containing supernatant $0.4 \times 10^6$ HEK293T cells were plated in 2 mL complete DMEM in 1 well of a 6-well dish. After 16 to 24 h, HEK293T cells were transfected with retroviral constructs encoding the gene of interest (2 μg per well), the retroviral packaging plasmids gag-pol (1 μg well), and VSV-G (100 ng/well) using GeneJuice transfection reagent (Novagen, 70967). Cells were incubated at 37˚C, 5% CO2 for approximately 12 h, and then the media was exchanged with RPMI containing 30% HI-FBS, and cells were incubated for another 36 h. After 36 h, the viral supernatant was collected using a 10-mL Luer-lock syringe attached to a blunt 18G needle and then filtered using a 0.45-mm filter unit into a 50-mL fal-con. The medium on target cells was removed, and viral supernatant was added to the cells at a 2:1 ratio with complete RPMI. Approximately 8 μg/ml polybrene was then added to the diluted virus-containing supernatant. The cells were then centrifuged at 800$g$ for 45 min at 37˚C, then harvested and plated in 24-well plates before being incubated for approximately 24 h at 37˚C, 5% CO2. Following incubation, the cells were collected by centrifugation, and the virus-containing medium was removed and replaced by complete RPMI. Transduced cells were passaged 3 times before frozen stocks were prepared. Cells were sorted for equal expression of Pyrin, and NLRP3 variants using fluorescence-assisted cell sorting on a FACS Aria cell sorter for equivalent expression of the fluorescent protein used as a marker of transduction.

## Flow cytometry analysis

THP1 cells were harvested and analyzed for mCherry expression using a MACSQuant YVB analyzer (Milltenyi Biotec) and the data analyzed using FlowJo.

## *C. difficile* coculture and supernatant generation

Experiments were performed with *C. difficile* DSM 28645 and DSM 29688 obtained from the German Collection of Microorganisms and Cell Cultures (DSMZ, Braunschweig, Germany). Main cultures were cultivated in RPMI 1640, supplemented 10% FBS, 0.014 mM iron sulfate, 4.16 mM cysteine, 4.33 mM proline, 1.11 mM valine, 1.12 mM leucine, 0.72 mM isoleucine, 0.22 mM tryptophan, 0.57 mM methionine, and 0.22 mM histidine at 2% $O_2$, 5% $CO_2$, 37˚C, 40% to 50% humidity using O2 Control InVitro Glove Box (Coy Labs, USA).

PBMCs were isolated from 3 different donors and differentiated into hMDM as described above. Two days before the experiment, cells were seeded in RPMI medium supplemented with 10% HI-FBS at $3.3 \times 10^5$ cells per well in 24-well plates and incubated at normoxic conditions at 37˚C for 24 h. On the following day, the cells were placed into a hypoxia chamber (2% $O_2$, 5% $CO_2$, 37˚C, 40% to 50% humidity) for another 24 h. On the same day, *C. difficile* main cultures of DSM 28645 (toxin-producing) and DSM 29688 (non-toxigenic) were inoculated at an $OD_{600nm}$ of approximately 0.01 and incubated for 24 h. On the day of the experiment, the medium was removed, and the cells were washed with 500 μl PBS. Following this, the cells were incubated with 375 μl RPMI or 375 μl RPMI containing 10 ng/ml LPS for 2 h. The

$OD_{600nm}$ of both *C. difficile* cultures was determined, and the number of bacterial density was determined using the following formulation:

$$C. \textit{difficile} \text{ per ml main culture} = 26,445,593 \times OD_{600nm}$$

The cells were centrifuged at 2,500*g* for 10 min. After centrifugation, the supernatant was passed through a 0.2-μM sterile filter. The pellet was resuspended in RPMI supplemented with 5 mM butyrate and with lower concentrations of glycine (reduced to 0.033 mM), cysteine (0 mM), proline (0 mM), isoleucine (0.095 mM), leucine (0.095 mM), methionine (0.026 mM), serine (0 mM), threonine (0.042 mM), and valine (0.042 mM), and the bacterial density was adjusted to a multiple of infection (MOI) of 300. The cells were treated with 375 μl sterile-filtered *C. difficile* supernatant, living *C. difficile* (300 MOI) or RPMI. The cells were additionally treated with 10 ng/ml LPS, 2 μM CP-456,773 or a combination of both. A lysis control was included by the addition of 0.5% triton-X100 in RPMI. After 3 or 6 h, the cell supernatant was collected, centrifuged for 10 min at 2,500*g*, and frozen at −80˚C. The cells were washed 2 times with 750 μl PBS and lysed by the addition of 80 μl M-PER with cOmplete Mini Protease Inhibitor for 5 min. The lysed cell suspensions were collected and stored at −80˚C.

## Caspase-1 activity assay

The caspase-1 activity assay was performed using the Caspase-Glo 1 inflammasome assay from Promega (G9951) as per the manufacturer's instructions. Briefly, cell-free supernatant from inflammasome-stimulated cells was mixed with equal amounts for reconstituted caspase-1 reagent and the Luminescence signal read on a SpectraMAX i3 (molecular devices) at 30, 60, and 90 min postmixing.

## qPCR

qPCR quantifications were performed essentially as previously described [42] with the following changes: 500 ng of RNA was used for the RT-PCR and the qPCR was performed using QuantStudio 6 PCR System (Thermo Fisher Scientific). The primer sequences were as follows: Hprt, forward 5′- TCAGGCAGTATAATCCAAAGATGGT-3′ and reverse 5′- AGTCTGGCTTATA TCCAACACTTCG-3′; *MEFV*, forward 5′- GGAAGGCCACCAGACACGG-3′ and reverse 5′- GTG CCCAGAAACTGCCTCGG-3′.

## Statistical analysis

All statistical analyses were performed using Prism GraphPad. The Student *t* test was used when the comparison was between 2 groups. A one-way ANOVA was used to analyze any experiment that included more than 2 groups.

## Supporting information

**S1 Data. The raw data underlying the graphs shown in Figs 1–7 and S1–S3.** Each tab contains the data from one figure, the data is labeled as in the graphs shown in the main figure. (XLSX)

**S1 Raw Material. Complete immunoblots for Figs 1–7 and S1–S3.** The excerpted portion of the immunoblot shown in the relevant figure is highlighted by a black box. (PDF)

**S1 Fig.** (**A**) Immunoblot of Rac glucosylation status in either monocytes or hMDM following treatment with the listed toxins (NXN variants lack glucosyltransferase activity).

Representative of 3 experiments. (**B**) Immunoblot of hMDM lysate sequentially probed with the α-Pyrin antibody preabsorbed against HEKs transfected with Pyrin, then with the α-Pyrin antibody preabsorbed against those transfected then empty vector (control). (**C**) Pyrin expression in monocytes or hMDM from 3 different donors. (**D**) Actin staining following incubation of monocytes or macrophages with or without LPS and TcdB. Treated cells were fixed and stained with Phalloidin 647 to detect actin (red) or with DAPI to detect nuclei (blue). White arrows highlight the changes in actin distribution between the 2 conditions. Images are representative from 3 separate donors. (**E**) IL-1β release from LPS-primed hMDM differentiated for 7 days in M-CSF and stimulated with either TcdA or TcdB +/ CP-456,773. Mean and SEM shown for 3 independent donors, $^* p < 0.05$, n.s. not significant. The underlying data can be found in the summary data file (S1 Data) in the tab S1E Fig.
(PDF)

**S2 Fig.** (**A**) Differentiated caspase-4-deficient BLaER1 cells were stimulated as in Fig 3A. IL-1β was assessed from the harvested supernatants. (**B**) LPS-primed differentiated WT BLaER1 cells were preincubated with TAK242 (2 μM, 30 min) then activated with TcdB (20 ng/ml) or nigericin (8 μM) for 2 h. Harvested supernatant was assessed for IL-1β. (**C**) Differentiated WT BLaER1 cells were preincubated with TAK242 then stimulated with LPS for 4 h. TNFα was assessed from the supernatant. (**D**) TNFα was measured for THP-1 cells from Fig 3G. Mean and SEM shown for 3 independent experiments. (**E**) LPS-primed (10 ng/ml, 3 h) human macrophages were treated either TcdB or the TcdB NXN mutant lacking glucosyltrasferase activity (20 ng/ml, 2.5 h). Supernatant was harvested and assessed for IL-1β or TNFα. For (**A**-**C**), the mean and SD of 3 technical replicates shown, representative of 3 independent experiments. For (**D**) and (**E**), the mean and SEM shown for 3 independent experiments. $^* p < 0.05$, n.s. not significant. The underlying data can be found in the summary data file (S1 Data) in the tab S2B–S2E Fig.
(PDF)

**S3 Fig.** (**A**) IFN-β (5,000 U/ml, 3 or 18 h) primed hMDM preincubated with compounds as previous and stimulated with TcdB (20 ng/ml) for 2.5 h. The supernatant was harvested and assessed for IL-18 release. (**B**) NLRP3 expression in either untreated hMDM or incubated with LPS for 3 h or 18 h as assessed by immunoblot, representative of 2 independent donors. Mean and SEM shown for 3 independent donors for cytokine release, $^* p < 0.05$, n.s. not significant. The underlying data can be found in the summary data file (S1 Data) in the tab S3A Fig.
(PDF)

## Acknowledgments

We thank C.C. Kolbe for setting up and optimizing the IL-18 Cytokine Bead Array assay and H. Tatge for purification of the different *C. difficile* toxins. We thank Dr. Steve Canham and Novartis for the gift of the BAA-473. We thank Niklas Mahnke for assistance with the FMF samples. We thank the Microscopy Core Facility and Flow Cytometry Core Facility at the University of Bonn for providing help, service and devices funded by the Deutsche Forschungsgemeinschaft (DFG–German Research Foundation)–Projektnummer 13123509.

## Author Contributions

**Conceptualization:** Matthew S. J. Mangan, Eicke Latz.

**Data curation:** Matthew S. J. Mangan.

**Formal analysis:** Matthew S. J. Mangan, Alexander Heinz, Florian I. Schmidt.

**Funding acquisition:** Ralf Gerhard, Eicke Latz.

**Investigation:** Matthew S. J. Mangan, Friederike Gorki, Karoline Krause, Alexander Heinz, Florian I. Schmidt.

**Methodology:** Matthew S. J. Mangan, Alexander Heinz.

**Resources:** Anne Pankow, Thomas Ebert, Dieter Jahn, Karsten Hiller, Veit Hornung, Marcus Maurer, Florian I. Schmidt, Ralf Gerhard.

**Writing – original draft:** Matthew S. J. Mangan, Eicke Latz.

**Writing – review & editing:** Matthew S. J. Mangan, Eicke Latz.

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
