## [Editor Report · Decision Letter 0]

22 Jun 2021

Dear Mathew,

Thank you for submitting your manuscript entitled "Clostridium difficile Toxin B activates the NLRP3 inflammasome in human macrophages, demonstrating a novel regulatory mechanism for the Pyrin inflammasome" for consideration as a Research Article by PLOS Biology. I have taken over the handling of your submission during my colleague Paula Jauregui's absence from the office this week, in order to prevent unnecessary loss of time.

Your manuscript has now been evaluated by the PLOS Biology editorial staff and I am writing to let you know that we would like to send your submission out for external peer review.

Please re-submit your manuscript within two working days, i.e. by Jun 24 2021 11:59PM.

Kind regards,

Nonia

Nonia Pariente, PhD 

Editor-in-Chief, PLOS Biology

npariente@plos.org

on behalf of

Paula Jauregui, PhD

Editor

PLOS Biology

---

## [Decision Letter · Decision Letter 1]

18 Aug 2021

Dear Dr. Mangan,

Thank you for submitting your manuscript "Clostridium difficile Toxin B activates the NLRP3 inflammasome in human macrophages, demonstrating a novel regulatory mechanism for the Pyrin inflammasome" for consideration as a Research Article at PLOS Biology. Your manuscript has been evaluated by the PLOS Biology editors, an Academic Editor with relevant expertise, and by several independent reviewers.

You will see that all the reviewers agree that the manuscript is interesting and it is an important finding, however, there are several issues that should be addressed. In particular, one of the main critiques is that the manuscript is light on depth. Reviewer 2 and 3 give you some suggestions on how to add insight to the paper (e.g., test other toxins, identifying minimal domain that activates NLRP3, etc.). Some of those proposed experiments will make the study more mechanistic. The reviewers also highlight some technical issues that should be solved. Please address all the reviewers issues.

In light of the reviews (below), we will not be able to accept the current version of the manuscript, but we would welcome re-submission of a much-revised version that takes into account the reviewers' comments. We cannot make any decision about publication until we have seen the revised manuscript and your response to the reviewers' comments. Your revised manuscript is also likely to be sent for further evaluation by the reviewers.

We expect to receive your revised manuscript within 3 months. 

**IMPORTANT - SUBMITTING YOUR REVISION**

*Re-submission Checklist*

*Published Peer Review*

*PLOS Data Policy*

*Blot and Gel Data Policy*

Sincerely,

Paula

---

Paula Jauregui, PhD

Associate Editor

PLOS Biology

REVIEWS:

Reviewer #1: Inflammasome responses during bacterial infections and auto-inflammatory syndromes.

Reviewer #2: Toxins and NLRP3 inflammasome.

Reviewer #3: Autoimmunity and Autoinflammation.

Reviewer #1: 

Mangan and colleagues described the interesting finding that in human macrophages (hMDMs) TcdB activates the NLRP3 inflammasome while both TcdA and TcdB fail to activate pyrin inflammasome while they do so in primary human monocytes. This is an important description since these two toxins are the main tools to activate the Pyrin inflammasome.

A catalytic mutant of TcdB, unable to modify RhoA (the key signal to trigger pyrin inflammasome activation) triggers NLRP3 activation. This is highly unexpected, nicely shown with the recombinant catalytic mutant TcdB protein! But not the focus of this manuscript.

Long LPS (or IFN) treatment induces pyrin expression and triggers TcdA/TcdB inflammasome responses that are pyrin-dependent. Several experimental systems are used to demonstrate that the transcriptional increase in Pyrin can restore TcdB-mediated pyrin inflammasome activation. Transcription regulation is clearly contributing to this regulation although other data in this manuscript suggest that the regulation is likely more complex.

Overall, while the mechanistic underlying the regulation of pyrin and the mechanisms associated with TcdB-mediated activation of NLRP3 remain to be deciphered, this paper is important to set the bases for future works aiming at understanding inflammasome regulation in healthy donors during C. difficile and potentially Yersinia or Burkholderia infections, and in patients with autoinflammatory diseases.

The knowledge on the Pyrin inflammasome is lagging far behind other inflammasomes and it is important for the field to gain novel insights onto this inflammasome complex which is highly relevant for infections and autoinflammatory syndromes.

At this stage a number of experiments are required to support in a robust manner the authors conclusions.

Main comments:

1-The most trivial explanation: difference in expression levels between primary monocyte and hMDM should be better established. Indeed, Pyrin has been described to have reduced expression level in hMDMs compared to monocytes both at the transcriptional and protein level (see Fig 2 in Gavrilin MA et al JI 2009).

The data is based here on one Western Blot panel with a single donor (Fig. 1C). Furthermore, Pyrin is migrating under 80kDa in this panel while it should be above. Pyrin can have multiple degradation bands and you should show the full Western blot (see Chae JJ et al Blood 2008 or your own WB Fig. 5D). 

This WB in Fig 1C thus needs to be repeated by showing the band corresponding to the Full Length protein and in three donors to robustly rule out that the expression level of the FL pyrin is not the trivial explanation. qRT-PCR must also be performed to validate (or not) the findings at the transcript level.

hMDMs are usually obtained after 7 (sometimes 10 days) of monocyte differentiation. If I am correct, you differentiate monocytes during 3 days. It would be important to validate the key findings (NLRP3-dependent TcdB response and lack of TcdA response) in hMDMs differentiated with a standard protocol.

2-The functionality of endogenous pyrin in BlaER1 cells and THP1 monocytes should be demonstrated if you would like to draw conclusions on the lack of role for endogenous pyrin in BlaER1 macrophages and THP1 macrophages. 

For WT BlaER1, Pyrin may be truncated since it migrates at a lower MW than upon complementation on the WB Fig 2E. Furthermore, pyrin up-regulation by LPS seems defective (please indicate the priming time and LPS concentration). Can you generate BlaER1 in a monocyte like differentiation to check the functionality of pyrin? Otherwise this limit should be clearly mentioned.

Same question for THP-1. Did you check the functionality of THP-1 pyrin inflammasome in THP-1 "monocyte" (i.e. in the absence of PMA and in response to TcdA and TcdB). It is described to be functional, but you should check it in your experimental system.

3- The manuscript ends on an unfinished Figure 6. Authors have investigated the step1 (S242 dephosphorylation which seems to be functional in hMDM), they thus need to investigate whether FMF patient hMDMs defective in the step 2 regulation would be responsive to TcdA as suggested by Shiba et al (JACI 2019) and whether TcdB responses would be blocked by colchicine, MCC950 or neither of each. 

Clearly, these experiments are missing. 

The legend in Figure 6 "and is disrupted by FMF mutations" suggests that the authors have at least thought about this experiment which must be performed in primary cells from FMF patients or at least using reconstitution of BlaERcells and/or THP-1 cells with FMF MEFV alleles.

4- We have learned from NLRP3 that inflammasome regulation is highly complex. While I have no doubt that MEFV up-regulation contributes to pyrin responses, this is likely not the only regulation as suggested by Fig 6 and Fig 1C. 

To strengthen the role of LPS treatment per se vs. up-regulation, Fig 5F should be performed without LPS treatment looking either at IL-18 or at ASC specks.

Fig 5F: blot is missing and should be shown. Do you really have MEFV overexpression. Based on Fig. 2E (overexpression in the MEFVko background), this is not obvious.

5- The discussion is extensive which is nice to see and read. One paper is missing and should be discussed Shiba T et al in JACI 2019 since it illustrates another difference between monocytes and hMDMs: the fact that in hMDMs from FMF patients, TcdA-mediated pyrin inflammasome activation is blocked by colchicine while it is not in monocytes. Furthermore, in this paper a short priming (2h) with a large dose of LPS (1ug/ml) is sufficient to promote TcdA -mediated pyrin inflammasome activation. Finally, looking carefully at the data, one might observe that hMDMs from healthy donors do not respond to TcdA while hMDMs from FMF patients do.

6-Numerous inaccuracies or overstatements (see minor comments).

Minor comments

1-I suggest to edit the title: demonstrating the presence of a novel regulatory mechanism. (The demonstration of the regulatory mechanism is not complete at the moment)

2-Clostridim is the old nomenclature, please edit throughout.

3-Line 55 and throughout: Please italicize gene name

4-Line 68: the redundancy is limited to one toxin

5-Line 75: pyrin oligomerization upon activation remains to be demonstrated (the current literature indicates that pyrin is a trimer at steady state).

6-Line 101: the requirement for microtubule dynamics (as opposed to microtubule polymerization/integrity) is controversial. It is based on paclitaxel (Taxol) shown to inhibit pyrin inflammasome activation in Gao et al but to be inefficient in Van Gorp et al and in Magnotti et al. 

7-Line 148: these results suggest that pyrin is held in an inactive state (instead of demonstrate)

8-Line 216: It would be interesting to assess whether RhoA is inhibited in hMDM by TcdA (and TcdB). The commercially available Rho A G-LISA activation assay could be used (although Fig 6A suggests Rho A is indeed inhibited). 

9-Line 238: since microtubules have been shown to contribute to NLRP3 activation (Magupalli VG et al Science 2020), the efficacy of colchicine suggests the dependence on pyrin but does not demonstrates it. 

10-Line 245: With the quality of the images at the moment, it is hard to conclude anything. We do see the cell death in monocytes, but the actin disruption in the presence of VX-765 is not obvious.

11-Line 394: change the title of figure 3: from "Pyrin is the responding sensor" to "NLRP3 is not the sensor". This is what is demonstrated in Fig 3.

12-Line 445, 453, 485, 532 : Pam3CSK4

13-Fig 4B: please edit the y-axis to MEFV/IL1b transcript or mRNA

14-Line 450: ISRE

15-Line 513: "demonstrating that increased expression is linked to pyrin reactivation."

This is an overstatement. At best, you have a correlation. Plus, how do you explain that you have lost the NLRP3 response?

16-Line 529: I am not sure you can compare a pool of untreated cells with a pool of siRNA-treated cells. You could have a full knock-down in 90% of the cells. 

17-Line 592: to my knowledge, no endogenous molecules have been demonstrated to inhibit RhoA signaling pathway and trigger pyrin activation.

18-Line 594: change "multiple" to "two"

19-Line 619: please double check Centola et al (ref 24) for the use of PBMCs vs. monocytes and Pam3CSK4 vs. LPS & TNF. Similarly, Pyrin is induced by F. novicida infection, a very poor TLR4 activator (Gavrilin A et al, JI 2009).

20-line 623: to inhibited

21-line 625: could you add a reference for inflammatory signals regulating AIM2 inflammasomes

22-line 631: Gram-negative

23-line 633: novidica

24-line 634: "Pyrin is the responding sensor to F. novicida in hMDM [25]." This is controversial see Lagrange et al Nat Com 2018 and Mitra S et al Plos 1, 2018.

25-line 637: "IL1b blocking therapy is only somewhat successful in preventing the disease [26]". 

This sentence conveys a false message. IL1b blocking therapy is successful (at least to my knowledge and according to ref 26).

26-Line 639 that stimulates

27-line 644: comparible

28-line 652: van Gorp et al did not demonstrate a requirement of the B30.2 domain for the colchicine sensitive pyrin activation. They demonstrated that FMF-associated point mutations in the B30.2 domain abolish colchicine-mediated inhibition, which is not the same (as described in the intro, mouse pyrin lacks B30.2 but its activation can be blocked by colchicine). 

29-line 668: my understanding was that AIM2 signaling was deficient in both human monocytes and macrophages not that there was a difference between monocytes and macrophages.

30-line 681: endogenous bile salts do not activate pyrin. Ref 29 describes potential bile acid metabolites which might be generated by the sequential action of two bacterial enzymes. 

31-line 689: here you could cite the Aim2 work by Gaidt and colleagues

32-line 693: I don't get the M-CSF point. Pyrin is highly expressed in neutrophils. 3 days of M-CSF treatment (Fig. 1C) does not modify Pyrin expression levels.

Reviewer #2: 

The manuscript by Mangan and colleagues entitled "Clostridium difficile Toxin B activates the NLRP3 inflammasome in human macrophages, demonstrating a novel regulatory mechanism for the Pyrin inflammasome" investigated the sensing of Toxin B by inflammasomes in human monocytes and macrophages. This is a very interesting topic and a timely study, but the results presented are not yet fully convincing. The main observation of the difference between monocytes and hMDM in sensing Toxin B is interesting but the molecular mechanism proposed is not clear. In addition, the results should include statistics for most panels and the discussion might be shortened.

Major points: 

1- It is not clear if the DSM28645 and DSM29688 strains used are isogenic? if not the authors should mention that the data shown Fig 1A are correlative and might be TcdA, TcdB dependent.

2- The finding that the TcdB activity is not required for the NLRP3 activation suggest it is recognized as a PAMP. It would be interesting to determine the minimal motif that would be sensed by NLRP3, and this motif should not be in TcdA. Expression of TcdB truncation in a reconstituted NLRP3 inflammasome in 293T (artificial but useful) model could be an option or in BLaER1 or THP-1. Another hypothesis is the activation of NLRP3 via a TcdB specific membrane receptor. In this receptor hypothesis, transfection of both TcdB or the TcdB NXN should not activate NLRP3.

3- To strengthen their conclusion the authors should investigate other RhoGTPase inhibiting toxins such as C3 or TecA that were shown to activate Pyrin through RhoA inhibition and determine if Pyrin or NLRP3 are required in this context.

4- In Figure S1A, the IL-1b secretion is only partially toxin dependent and only partially NLRP3 dependent and IL-18 is increased in TcdA/B-? this is confusing compared to the Fig1A and the results with the supernatant. Only 3 points are visible in the graph. Is this statistically relevant or more replicates are needed to conclude? 

5- It is not clear if the authors are using ASC-BFP transduced WT BLaER1 cells as indicated in the fig legend of Fig2B or BLaER1 cell line overexpressing ASC-mCherry as indicated in the main text. 

6- The figure 2C, 2D, 5F should mention the fact that cells used are NLRP3-Casp4 KO 

7- In the Figure 2D and 2F, why the authors analyzed the TNFa only in LPS treated cells and not with all the sample shown for IL-1b, they should at least measure it for samples with TcdB that is the focus of the paper?

8- A TNFa measurement in Fig S2E would allow to determine if the difference observed between Tcdb and Tcdb NxN mutant is specific to the NLRP3 inflammasome in hMDM

9- A complementation in mefv ko (BLaER1 or THP1) of the human and murine forms of Pyrin could provide information.

10- In Figure S2D, the authors statement: "Furthermore, the different KO lines secreted similar amounts of TNFa in response to LPS" is rather confusing without statistics. 

11- The main hypothesis of the difference in terms of Pyrin response rely on the expression level of Pyrin in hMDM versus Monocytes. The authors should show a comparison between these 2 cell types in terms of Pyrin expression level after 18h of LPS (as in Fig 6). This is important considering that at 3h LPS Pyrin levels appears to be similar in both cell types in Fig 1C. 

12- The discussion mentioned post-translational modifications that could explain similar level of protein but different activities. This could be tested using Pyrin IP followed by WB using phospho-antibodies, anti-Ub-antibodies, 2D gels or mass spec analysis.

Minor points :

1- The introduction should mention the previous work on the NLRP3 sensing of RhoGTPase targeting toxins.

2- The Rac WB shown in Sup Fig-1 indicated the activity of both toxins. The actin blot is missing in Sup Fig1b monocytes and in hMDM the actin is not visible, and the Rac1/2/3 blot is missing.

3- In Figure 1E the actin loading control is missing.

4- P 18 line 431 this sentence is not clear concerning IL-1β « Notably, only LPS increased Pyrin expression after 5h, while LPS, IFN-β and IFN-γ but not TNFa, IL-1β or Pam3CS4K increased expression after 18h (Fig. 4a) »

5- Unless I missed it, the title of Fig 6 refers to a disrupted signal in FMF mutations that is not shown "The B30.2 domain regulates Pyrin activation in human macrophages, and is disrupted by FMF mutations"

Reviewer #3: In this study Mangan et al. investigate the inflammasome response to C. difficile toxins TcdA and B in human monocyte-derived macrophages (hMDM). The authors find that in contrast to monocytes, TcdA and TcdB do not signal through the Pyrin inflammasome in hMDM. Rather only TcdB, not TcdA activates the NLRP3 inflammasome in these cells. This latter aspect is not entirely novel given prior literature showing that TcdA/B activates NLRP3 in differentiated THP-1 macrophages however the difference observed between the two toxins in hMDM is intriguing. The authors propose that prolonged (18h) exposure to LPS or type 1 IFN increases Pyrin expression in hMDMs and reactivates engagement of the Pyrin inflammasome in these cells. Strengths of the study include a well-written and well-discussed paper, and the different macrophage model systems employed including primary cells and cell lines. Weaknesses are a striking lack of mechanistic insight relating to the central findings of the paper, especially - 1) how Pyrin is held in an inactive state when monocytes differentiate to macrophages despite no change in actual expression of Pyrin between these cell types, 2) how priming relieves this inhibition and makes Pyrin conducive to being activated, and 3) how TcdB activates NLRP3 in hMDM, although on this point the authors have tried to rule out the conventional mechanisms by which TcdB works to activate the Pyrin inflammasome. There are also some technical problems with blots, images and missing data. The basic finding of the paper is interesting however there are several issues that need to be addressed.

Major comments:

1. Is K+ efflux required for NLRP3 activation by TcdB in hMDM? This is a common mechanism of NLRP3 activation by toxins and is easily tested in presence of increasing concentrations of K+.

2. It is appreciated that the authors have shown independent donors on graphs, however given the spread of the data statistics are needed throughout the paper to be able to better appreciate which differences are significant and which aren't. Fig. 5b, 5e, S2a are some examples of where this is especially difficult.

3. Fig. S1a: Unlike IL-1b, there is no IL-18 with the TcdA+/B+ strain so IL-18 produced upon infection with C. difficile not seem to be dependent on TcdA/B. However the text (lines 173-174) claims that cytokine release was "primarily reliant on toxin expression when cells were infected directly". This needs to be explained / clarified. The requirement of a LPS priming signal for IL-18 which is thought to be pre-formed is also puzzling.

4. Please show better blots and loading controls. a) Fig. S1B - In the blots on the left, missing band with Rac1/2/3 polyclonal (presumably a loading control) in the TcdB lane makes it difficult to appreciate if lack of a band in the upper blot is due to glucosylation or a loading issue. Similarly b-actin loading control in the blots on the right is hardly visible. b) Fig. 1E - Loading controls on cells lysates are needed alongside the IL-1b and caspase-1 blots.

5. Fig. S1C: Disrupted actin is not clear or visible in most images. Better resolution / higher magnification images and insets are needed to convincingly appreciate the impact of the toxin on actin disruption.

6. Fig 2a: Baseline caspase-1 activity in BLaER1 cells is quite high. From the data presented, it seems likely that the differences in caspase-1 activity seen with TcdB in presence and absence of the NLRP3 inhibitor will likely not hold when these baselines are accounted for. A blot for caspase-1 is needed to convincingly show if TcdB mediated caspase-1 activation in these cells is sensitive to NLRP3 inhibition. 

7. It is rather difficult to see ASC specks in TcdB treated cells in Fig. 2b. Higher magnification images and insets are needed in this figure. 

8. To conclude that caspase-4 is not required for the TcdB response the authors would need to compare WT and Caspase-4 KO BLaER1 cells side by side in the same experiment (Fig S2a). 

9. There seem to be differences in TNF responses between the different THP1 lines in Fig S2d contrary to what is stated in the text. Please explain. Statistics are needed to test if these differences are indeed significant.

10. Fig. 3 has no corroborating data showing that Pyrin is the sensor that responds to TcdB in mouse BMDMs under the experimental conditions used by the authors. Although I appreciate that this has been previously reported, some form of data (preferably using WT and Pyrin KO BMDMs) which shows that this holds up for the authors would strengthen the key message of this figure. 

11. Figs. 3c and 3d - please show better blots and perform loading controls on the cell lysates. The TcdB bands are too dim to fully appreciate differences. In 3d, active caspase-1 in response to dA:dT appears to be less in presence of NLRP3 inhibitor. Without loading controls it is quite difficult to appreciate what is real and what is an artifact. 

12. The text and figure legend mentions that IL-b was also assessed in Figs 3c and 3d, however IL-1b data is missing from these figures.

13. The data presented do not convince me that expression of Pyrin is the limiting factor for its activation in hMDM. The terminology "necessary and sufficient" is a bit of an overstatement in this regard. Expression of Pyrin protein in monocytes and hMDM is equivalent (Fig. 1c) but hMDM still do not trigger a Pyrin response in the absence of prolonged priming. Because all experiments in Fig. 5 are performed under the cover a LPS or IFN-b priming signal, one likely possibility is that a post-translational modification induced by priming is the trigger for Pyrin reactivation. Alternatively, it is possible that a decrease in NLRP3 upon prolonged priming of hMDM reveals a role for Pyrin in these cells. This latter possibility needs to be tested and the authors should reword to text to discuss these alternatives.

14. Related to the above comment, does NLRP3 expression go down upon prolonged priming (18h) thereby making Pyrin the preferred sensor in hMDM? The authors should assess NLRP3 expression at 5 and 18h (Fig. 4). This is important because NLRP3 expression is also induced by LPS and some of the other stimuli tested by the authors. 

15. Fig 5E - The decrease in IL-1b in TcdA treated cells in presence of the TRIF inhibitor is not convincing and seems modest at best. Is this difference significant?

16. Fig 6 is incomplete. The title of the figure and discussion section imply that there is data relating to the B30.2 domain and FMF mutations in this figure but these data are not present in the paper.

---

## [Decision Letter · Decision Letter 2]

9 Jun 2022

Dear Dr Mangan,

Thank you for your patience while we considered your revised manuscript "­­­Pyrin inflammasome activation in human macrophages requires transcriptional licensing, which is deregulated in Familial Mediterranean Fever, revealing a role for the NLRP3 inflammasome in sensing Clostridioides difficile Toxin B" for publication as a Research Article at PLOS Biology. Your revised study has been evaluated by the PLOS Biology editors, the Academic Editor and the original reviewers. Please accept my sincere apologies again for the delays in sending this editorial decision.

In light of the reviews, which you will find at the end of this email, we would like to invite you to revise the work to thoroughly address the reviewers' reports. As you will see below, the reviewers have noted improvements on the previous version of your manuscript; however, they still raise concerns with the interpretation of the results and the strength of the conclusions. Having discussed the reports with the academic editor, whereas we agree that the investigation of the mechanisms of NLRP3 activation is beyond the scope of your current study, we also feel that the concerns raised around the data presented in Figure 7 are important, and therefore need to be carefully addressed. In this regard, reviewer 3 in point #7 offers suggestions on how to address experimentally the conclusiveness of Figure 7 data. 

Given the extent of revision needed, we cannot make a decision about publication until we have seen the revised manuscript and your response to the reviewers' comments. Your revised manuscript is likely to be sent for further evaluation by all or a subset of the reviewers.

**IMPORTANT - SUBMITTING YOUR REVISION**

*Re-submission Checklist*

*Published Peer Review*

*PLOS Data Policy*

*Blot and Gel Data Policy*

Sincerely,

Dario

Dario Ummarino, PhD, 

Senior Editor

PLOS Biology

dummarino@plos.org

REVIEWS:

Reviewer #1: 

The authors have strengthened their data by including more healthy controls, by checking the specificity of the pyrin antibody, adding another pyrin stimulus (BAA473) and completing the missing part on FMF patients.

While I am still convinced that this is an important story for the field, the part on FMF patient is not robust enough at this stage to be presented (at least in regards to the third panel related to TcdB). The experiments are presented with 2 to 3 patients which would fine if the patients were typical FMF patients (in terms of MEFV mutations). Yet, some of the patients have very peculiar genotypes (controversial in terms of pathogenicity) precluding to draw any conclusions on FMF patients (see major point below). 

All the other points are minor points to improve the details of the manuscript. 

I agree with the authors that the mechanisms underlying the recognition of TcdB by the NLRP3 inflammasome is out of the scope of the current manuscript (although super interesting).

Major point:

-The FMF data are impressive and very striking for the first 2 panels of Fig. 7D despite the use of only 2 to 3 patients. The data presenting in Fig 7D (panel 1 and 2) is crystal clear in regards 

-to the colchicine-independence of TcdA in FMF monocytes (Van Gorp et al)

-to the response to TcdA in hMDM from FMF monocytes (Shiba et al) (Honda et al)

-to the colchicine-dependence of TcdA in hMDM from FMF patients (Shiba et al)(Honda et al).

Unfortunately, this is not the case for the third panel of Fig. 7D (the TcdB experiment in hMDM) which is also performed with only two FMF patients, the first one compound heterozygous for E148Q/A744S, the second one compound heterozygous for M694V/A744S. The very peculiar genotypes of these patients may explain why macrophages from these 2 FMF patients behave as macrophages from HD in regards to TcdB.

Indeed, the E148Q mutation is largely recognized as a non-pathogenic mutation (see Van Gorp H et al. Ann Rheum Dis 2020, Honda Y et al J. Clin Immunol 2021). 

The A744S mutation is a variant of unknown significance whose pathogenicity is also doubtful (see Alsubaie L et al. Annals of human genetics 2019 and Honda Y et al J. Clin Immunol 2021).

In addition to the very atypical/controversial genotypes, one patient barely responded to TcdB. 

Altogether, at this stage, you can not conclude on TcdB and FMF patients based on these two patients. Other patients -preferably homozygous or compound heterozygous with two pathogenic mutations should be included. 

While repeating this experiment I suggest adding a condition where you add both colchicine and CP to see whether the NLRP3 response may hide a pyrin response. 

This is very important since at the moment, the data suggest that the TcdB response in FMF hMDM is still fully CP/NLRP3-dependent and colchicine-independent suggesting that the TcdB licensing step is still required in FMF patients hMDMs (In contrast to what stated in the abstract). This might be true but cannot be concluded with these two FMF patients with atypical mutations.

The authors should be able to rapidly obtain blood from 2-3 FMF patients with classical mutations (e.g. homozygous M694V) to finalize figure 7 and the response to TcdB in FMF patients. This is a key experiment for the present manuscript and needs to be performed to sustain the authors conclusions (or present the complexity of the pyrin infammasome regulation).

Minor points:

1-Line 140-147 (from the track change version): The pyrin-dependent signaling of TcdA/ TcdB in BMDM is well demonstrated including with genetic demonstration (see van gorp et al. PNAS 2016, Fig 1 and Fig S1 fro both TcdA and TcdB, see Fig 1& 3,4 and extended fig1 in Xu H et al Nature 2014 for TcdB

2-Line 348-349: since BlaER1 cells are typically considered as monocyte/macrophage and since you did not check whether THP-1 monocytes are pyrin-responsive, please change the Figure 3 title to "NLRP3, not pyrin, is the responding inflammasome sensor to TcdB in human macrophages BlaER1 cells and THP-1 macrophages."

Also consistently use the terminology monocytes/macrophages (or cells) for the BlaER1 cells instead of macrophages (e.g. line 941)

3-Line 408: please rephrase: "these cells that they still released..."

4-Line 410-Fig S2c?+ correct line 429

5-line 667: please edit "more pyrin than"

6-line 700: please correct S208 instead of S204

7-Line 744: please check figure numbers and letters

8-The experiments with BAA-473 (Fig. 7B) does not make sense based on your current model. You are overexpressing pyrin hence you should have an active pyrin inflammasome in human cells. This should be discussed.

9-line 750: Replace "demonstrating that it is likely" by "suggesting" and line 761 "indicated" by "suggested" (murine pyrin differs in several aspects from human pyrin including but not limited to, the lack of the B30.2 domain).

10-line 906-910. This is inaccurate see Omenetti A et al ARD 2014, Jamilloux Y et al. Rheumatology 2017, Honda Y et al J. Clin Immunol 2021

11-Line 937-938: this alternative hypothesis does not make sense since when you block pyrin with colchicine, the pyrin pathway cannot predominate and yet you still do not have NLRP3 activation.

12-Line 951; you are right it has not been tested in hMDM (although AIM2 responds to T. gondii in hMDM to activate caspase-8 Fisch D et al. EMBO 2019) but it has been tested in human BMDM (Gaidt MM et al, cell 2017), so I am not convinced that Aim2 can be used as an example to compare monocytes and macrophages (in contrast to the species specificity see lines 978).

Reviewer #2: The authors greatly improved both the quality and the interest of their manuscript clarifying and adding new data including with FMF patient samples. Nevertheless, I still have 2 minor points regarding an unanswered question and the new added data. 

Minor points:

1- Unless I missed it, I was not able to find the comment in the introduction referring to the recently described role of NLRP3 in sensing toxins and virulence factors targeting Rho GTPases. Please provide line and page number.

2- The new figure 7 is a major addition to the work. I would be recomend to increase the number of samples and to perform statistics.

Reviewer #3: Mangan et al present a revised version of their paper. The authors have adequately addressed some of my comments and furnished missing data, however issues remain. I would request the authors to pay attention to detail in their submission as many figure panels in the reviewer responses (in elsewhere in the text) are mis-cited making it quite tedious for the reader to follow the paper.

Major issues:

1. It is unclear what is being referred to as disrupted actin cytoskeleton in the images in Fig. S1D. Pointing this out with arrowheads would be useful.

2. Fig. S1B: Unclear what the difference between the two lanes on the blots is. Please label them. 

3. Lines 238-239: '…found that Pyrin expression was comparable in hMDM and monocytes (Fig. 1c, Fig S1c)." This is clearly not the case. Pyrin expression in monocytes is less that in than hMDM (Fig. 1C, S1B, S1C) even though pyrin isn't being activated in hMDM. The text needs to state so. 

4. Fig. S3A: The response of both 3h and 18h IFN-beta primed cells to TcdB is reduced with colchicine. This needs to be explained / clarified as isn't it only the 18h Pyrin mediated IL-18 that you expect to be sensitive to colchicine? 

5. Fig S3 legend needs to be fixed. 

6. Fig. 3b: please indicate which color on the images corresponds to what either in the figure or in the legend. 

7. Data in Fig. 7 on the B30.2 domain of human pyrin in inhibiting its activation are inconclusive. In order to claim an inhibitory role for the B30.2 domain in activation of human pyrin the authors would need to delete this domain in human pyrin and show the data alongside full length human pyrin in Fig. 7B. A comparison with mouse pyrin is not sufficient as there could be many other factors, sequence dissimilarities etc contributing to differences in pyrin activation between these two species. The response of human pyrin without the B30.2 domain to TcdA and BAA-473 needs to be tested to reach useful conclusions.

8. I am a bit confused by the data in Fig. 7B where the authors show that BAA-473 cannot activate human pyrin even upon its overexpression. This runs contrary to the data in human MDMs in the rest of the paper (Fig. 6A for example) where it is implied that induction of pyrin expression by prolonged exposure to LPS can lead to its subsequent activation by BAA-473. 

9. In Fig. 7D the same FMF donors/mutations are not assessed for TcdA and TcdB responses. Different mutations can act in different ways making it difficult to appreciate any conclusions drawn from the middle and right (hMDM) panels in 7D. 

10. Donor numbers for which the western blot is shown in Fig. 7E are not the same as those used for assessing functional responses in 7D. A western blot for NLRP3 is also needed alongside the pyrin and actin blots in Fig. 7E to appreciate whether or not the NLRP3 sensitive effects claimed in Fig. 7D rightmost panel are related to differences in NLRP3 expression among patients. 

11. It is unclear why are FMF cells in Fig. 7D primed with P3C? Why prime these cells at all? The point being made in the discussion (line 791-794) and elsewhere in text is that these mutations enable pyrin activation in hMDM 'without any requirement for priming or overexpression.'

12. Lines 495-498: please rephrase as the language currently implies that LPS and P3C can cause an increase in IL-1b but not Pyrin. This is not correct as LPS causes an increase in Pyrin. 

13. Line 491: IL-1b is not tested in Fig. 5a and its mention should be deleted from the text.

14. Line 482: Fix the sentence. Do you mean "…whether inflammatory conditions increased the expression of Pyrin and thus potentially enable its activation"?

15. Line 475: change text to "may depend on Pyrin" and include the citation which showed that the response to TcdA/B in murine macrophages can depend on pyrin. 

16. Line 369: Wrong fig panel is cited. It should be Fig. S2C. Also fix the figure legend of Fig. S2 as it is inconsistent with the figure. Panels in this figure are wrongly cited in the text at other locations too.

17. Figure legend of Fig. S3 is inconsistent with the figure and needs to be fixed.

18. Lines 769-772: 'Whilst pyrin overexpression….' This sentence is confusing and it is unclear what point is being made so in the interest of clarity it may be best to delete it. 

19. Lines 745-747: 'Pyrin activation could be established for all stimuli by prolonged stimulation with either LPS or type I or II interferon….' This needs to be rephrased as pyrin 'activation' upon priming with type II IFN is not shown in the paper. Only an induction of pyrin expression is shown. 

20. Line 632: "However, in the experiment carried out in the Pyrin KO BLaER1 cells we did not determine….". This sentence is quite confusing and can be rephrased quite simply to indicate that the prior experiment was carried out in NLRP3 sufficient BLaER1 cells and Pyrin reconstitution in those cells was closer to baseline as compared the vast overexpression of Pyrin seen in Fig. 6F. 

21. Incorrect figure panels are cited at several places:

Line 729: should be Fig. 7e

Lines 717, 720 and 723: should be panel 7d

Lines 699 and 701: should be Fig. 7b

Line 691: should be Fig. 7c

Minor issues:

Line 1153: 'Following, the cells…' fix this sentence.

Line 1141: tryptophan

Line 1120-1122: 'The cells were then centrifuged at 800g…' Please check that there aren't inaccuracies in this sentence. It seems unusual that cells were harvested and plated in 24 well plates after centrifugation with virus + polybrene. Usually cells are not disturbed and are directly incubated after the centrifugation step to allow virus infection to proceed. 

Line 1090: Following treatment, "cells" were… Fix the sentence.

Line 1000: fix the sentence. Vx-765 "for" 30 min….

Line 982: "where" applicable

Lines 984-986: Fix sentences. Precoated "with" poly l lysine…; removed and "replaced with" fresh media…

Line 968: generate

Line 966: manufacturer's

Line 955: immunoblot

Fig. S1C, line 1366: four different donors should be 'two' different donors.

Line 885: irrelevant citation

Lin 745: steady state

Line 648: dephosphorylation

Line 616: …more Pyrin 'than' the control…

Line 607: 'prevent' Pyrin activation should be changed to 'decrease' or 'reduce'.

---

## [Editor Report · Decision Letter 3]

12 Sep 2022

Dear Dr Mangan,

Thank you for your patience while we considered your revised manuscript "Transcriptional licensing is required for­­­ Pyrin inflammasome activation in human macrophages, which is bypassed by mutations causing Familial Mediterranean Fever" for publication as a Research Article at PLOS Biology. This revised version of your manuscript has been evaluated by the PLOS Biology editors and the Academic Editor.

Based on our Academic Editor's assessment of your revision, we are likely to accept this manuscript for publication, provided you satisfactorily address the following data and other policy-related requests.

1. ETHICS STATEMENT:

**Please include information about the form of consent (written/oral) given for research involving human participants. **All research involving human participants must have been approved by the authors' Institutional Review Board (IRB) or an equivalent committee, and all clinical investigation must have been conducted according to the principles expressed in the Declaration of Helsinki.

2. DATA POLICY:

A) Supplementary files (e.g., excel). Please ensure that all data files are uploaded as 'Supporting Information' and are invariably referred to (in the manuscript, figure legends, and the Description field when uploading your files) using the following format verbatim: S1 Data, S2 Data, etc. Multiple panels of a single or even several figures can be included as multiple sheets in one excel file that is saved using exactly the following convention: S1_Data.xlsx (using an underscore).

B) Deposition in a publicly available repository. Please also provide the accession code or a reviewer link so that we may view your data before publication.

Regardless of the method selected, please ensure that you provide the individual numerical values that underlie the summary data displayed in the following figure panels as they are essential for readers to assess your analysis and to reproduce it: Figures 1ABD, 2AB, 3ABDFG, 4AB, 5B, 6ABCDEGH, 7CEFG, and Supplementary figures S1E, S2ABCDE, S3A.

**Please also ensure that figure legends in your manuscript include information on where the underlying data can be found, and ensure your supplemental data file/s has a legend.**

3. Please provide size bars for the microscopy pictures in figure 3B and supplementary figure S1D.

4. We suggest a change in the title: "Transcriptional licensing is required for ­­­Pyrin inflammasome activation in human macrophages and bypassed by mutations causing Familial Mediterranean Fever".

We expect to receive your revised manuscript within two weeks.

*Published Peer Review History*

*Press*

Sincerely,

Paula

Senior Editor,

pjaureguionieva@plos.org,

PLOS Biology

---

## [Editor Report · Decision Letter 4]

23 Sep 2022

Dear Dr. Mangan,

Thank you for the submission of your revised Research Article "Transcriptional licensing is required for Pyrin inflammasome activation in human macrophages and bypassed by mutations causing Familial Mediterranean Fever" for publication in PLOS Biology. On behalf of my colleagues and the Academic Editor, Ken Cadwell, I am pleased to say that we can in principle accept your manuscript for publication, provided you address any remaining formatting and reporting issues. These will be detailed in an email you should receive within 2-3 business days from our colleagues in the journal operations team; no action is required from you until then. Please note that we will not be able to formally accept your manuscript and schedule it for publication until you have completed any requested changes.

PRESS

Sincerely, 

Paula

---

Senior Editor

PLOS Biology
